

# Integrable Floquet QFT:
# Elasticity and factorization under periodic driving

**Axel Cortés Cubero**[1*]

**1** Institute for Theoretical Physics, Center for Extreme Matter and Emergent Phenomena,
Utrecht University, Princetonplein 5, 3584 CC Utrecht, the Netherlands

* a.cortescubero@uu.nl

## Abstract

In (1+1)-dimensional quantum field theory, integrability is typically defined as the existence of an infinite number of local charges of different Lorentz spin, which commute with the Hamiltonian. A well known consequence of integrability is that scattering of particles is elastic and factorizable. These properties are the basis for the bootstrap program, which leads to the exact computation of S-matrices and form factors. We consider periodically-driven field theories, whose stroboscopic time-evolution is described by a Floquet Hamiltonian. It was recently proposed by Gritsev and Polkovnikov that it is possible for some form of integrability to be preserved even in driven systems. If a driving protocol exists such that the Floquet Hamiltonian is integrable (such that there is an infinite number of local and independent charges, a subset of which are parity-even, that commute with it), we show that there are strong conditions on the stroboscopic time evolution of particle trajectories, analogous to S-matrix elasticity and factorization. We propose a new set of axioms for the time evolution of particles which outline a new bootstrap program, which can be used to identify and classify integrable Floquet protocols. We present some simple examples of driving protocols where Floquet integrability is manifest; in particular, we also show that under certain conditions, some integrable protocols proposed by Gritsev and Polkovnikov are solutions of our new bootstrap equations.



# 1   Introduction

The notion of integrability is simply defined in classical many-body systems, as the existence of as many conserved quantities as degrees of freedom. In these cases the classical equations of motions can be integrated and the dynamics are exactly solvable ( [1, 2] and references therein).

Defining integrability in quantum many-body systems is a more subtle issue (see for instance [3]), especially in systems that do not have a direct classical analogue, such as quantum spin chains, or fermionic systems, where the concept of number of degrees of freedom is not well defined. Furthermore, in quantum field theory, there should be an infinite continuum of degrees of freedom, so it is initially unclear what is the structure of conserved quantities needed to lead to exact solvability.

In (1+1)-dimensional relativistic quantum field theory (QFT), exact solvability is associated with the properties of elasticity and factorization of the S-matrix. From these properties, combined with constraints such as Lorentz invariance and unitarity, it is possible to write a set of nontrivial axioms, which enable the exact computation of the S-matrix [4]. It is also possible in a similar manner to find exact analytic expressions for matrix elements of local operators, by requiring that these satisfy a set of nontrivial axioms. Once these matrix elements are known, it is possible to write analytic expressions for correlation functions of these local operators [5–8].

The pragmatic approach to defining integrability in QFT is then to look for a model which has enough conserved quantities, such that the properties of elasticity and factorization arise. It is known that for relativistic QFT's, elasticity and factorization are guaranteed by the existence of an infinite discrete set of local conserved charges, with different integer values of Lorentz spin [1]. We will review the details of these conserved charges, and their connection to elasticity and factorization in the next section.

---

[1]It has been recently proposed that this discrete set of charges, while guaranteeing elasticity and factorization, is not sufficient to define a complete generalized Gibbs ensemble, which arise in certain non-equilibrium problems [9–12]. Extensions to the standard set of discrete local conserved charges in QFT have been proposed in [13–16].

The properties of elasticity and factorization can be used to define new integrable quantum field theories in a self consistent manner, often without use or knowledge of the model's Hamiltonian [17]. This is done by simply searching for an S-matrix that satisfies all the necessary axioms, which are consistent with knowledge of the spectrum of particles and symmetry properties of the system. Following this program, one can compute physical quantities such as correlation functions, and thermal partition functions [18] without use of any Hamiltonian or Lagrangian formalism.

It was recently proposed by Gritsev and Polkovnikov in [19], that it is possible to search for integrability, not only in equilibrium quantum many body systems, but also in periodically driven systems. Just as there was difficulty defining the meaning of integrability in an equilibrium quantum system, as compared to classical integrability, it is again not immediately clear how integrability should be defined in a driven quantum system.

Several ways to define integrability in driven systems were discussed in [19]. A significant object to consider is the so-called Floquet Hamiltonian, which describes the stroboscopic time evolution of the system (which we will discuss in more detail in Section 3). One can then define integrability by searching for Floquet Hamiltonians with special properties that can lead to some exact solvability. Different properties of an integrable Floquet Hamiltonian were proposed, such as Floquet Hamiltonians which commute with an infinite number of local operators, or whose energy level statistics show level crossing, in analogy to equilibrium integrable systems.

The main property of integrability displayed by all the examples presented in [19] is that the entropy of the system does not increase upon periodic driving, or the system does not "heat up". This is an important distinction because it has been argued that periodic driving generally leads to an increase in temperature in many-body quantum systems [20, 21]. The absence of heating has also been studied in driven systems exhibiting many-body localization [22–26].

Besides discussing various definitions of integrability in periodically driven systems, we argue that it is not clear from the results of [19] if and how, this form of integrability can lead to any exact solvability. In other words, one can identify if a driven system is integrable, but it is not clear what is the computational advantage of it being integrable.

Our approach in this paper will be similar to the standard equilibrium approach to integrable QFT's. We will focus on what are the desired properties of integrability that may lead to exact solutions. In equilibrium, these properties are elasticity and factorization, and an integrable system is one that has these properties. For a driven system governed by a Floquet Hamiltonian, we define integrability as the existence of some set of charges which commute with the Floquet Hamiltonian, which ensure some properties analogous to elasticity and factorization. The necessary structure of these conserved quantities, and the applications towards analytic computations will be discussed in detail in Section 3.

Once we have established that a Floquet Hamiltonian, describing a driven QFT, is integrable, providing some simple assumptions about the driving protocol, we are able to write down a set of axioms, which greatly constrain the stroboscopic time evolution of the system. These axioms are analogous to the S-matrix axioms in equilibrium QFT. We propose these new axioms can be used to identify and classify new integrable driving protocols in QFT.

We point out that the program followed in this paper is also similar to how integrability was applied to QFT's with a boundary, in [27]. In that case, it was found that it is sometimes possible to implement some special boundary conditions, such that there are still an infinite number of conserved charges which commute with the Hamiltonian describing the QFT with a boundary. From this, it follows that elasticity and factorization still apply in the boundary theory, and scattering of a particle against a boundary is described by an elastic reflection matrix. From elasticity and factorization, a set of axioms can be formulated which greatly constrain the form of the reflection matrix. One can then classify the different possible integrable boundary

conditions, by searching for reflection matrices which are solutions to the set of axioms.

In the following section we will review the standard approach to integrability in equilibrium relativistic QFT, and show the axioms that the S-matrix satisfies. In Section 3, we introduce the concept of periodically driven QFT's, and derive what are the consequences of having an integrable Floquet Hamiltonian, which are analogous to elasticity and factorization. In Section 4, we show how these properties can be used to define a useful set of axioms constraining the stroboscopic time evolution.

In Section 5, we study two simple driving protocols, where we periodically change the value of the mass in a free bosonic and a free fermionic QFT. We show that even in these simple free-theory protocols, the resulting Floquet Hamiltonians are not integrable. Nevertheless, the computational tools developed for this problem can help us to identify some other simple integrable driving protocols. In Section 6, we show that there exist some simple examples of driving protocols which exhibit integrability, and we show how each of these processes is a solution to the set of axioms defined in Section 4. In particular we study the proposed integrable protocols presented in [19], and we discuss in which cases our definitions of integrability agree with each other.

## 2 Integrable QFT and the S-matrix bootstrap program

We will now show a brief review of some useful consequences of integrability in QFT. In particular, we discuss how the presence of an infinite number of conserved charges implies elastic and factorized scattering. We then show a few axioms that follow from these properties, which restrict the form of the S-matrix. The arguments shown in this section closely parallel those presented in [28], which is a deeper and more extensive review on this subject.

The spectrum of a relativistic quantum field theory is characterized by particle excitations which satisfy the dispersion relation $E^2 = (mc^2)^2 + (pc)^2$, where $E$, $p$ and $m$ are the particle's energy, momentum and mass, respectively, and $c$ is the speed of light. We will set $c = 1$ from now on. In (1+1)-dimensional systems, this dispersion can be parametrized as

$$E = m \cosh \theta, \quad p = m \sinh \theta, \tag{1}$$

where $\theta$ is the particle rapidity.

The total energy or momentum of a state is given by the sum of contributions (1) for each excited particle. For some multi-particle state, $|\{\theta_i\}\rangle$, where $\{\theta_i\}$ is some set of particle rapidities, we have

$$H|\{\theta_i\}\rangle = \sum_i m \cosh \theta_i |\{\theta_i\}\rangle, \quad P|\{\theta_i\}\rangle = \sum_i m \sinh \theta_i |\{\theta_i\}\rangle.$$

The fact that the total energy and momentum can be separated into contributions from each individual particle is a consequence of the locality of the Hamiltonian and momentum operators. For a nonlocal operator, one can expect that contributions from distant particles do not separate.

We point out that the Hamiltonian and momentum operators transform as Lorentz vectors, *i.e.*, spin-1 operators. This can be seen more clearly by writing these operators in terms of light-cone components

$$P^\pm |\{\theta_i\}\rangle \equiv H \pm P|\{\theta_i\}\rangle = \sum_i m \, e^{\pm \theta_i} |\{\theta_i\}\rangle.$$

Under a Lorentz boost, the rapidity of all particles in a state is shifted by some constant as $\{\theta_i\} \to \{\theta_i + \alpha\}$, and the light-cone momenta of each particle transforms as $p^\pm(\theta_i) \to e^{\pm \alpha} p^\pm(\theta_i)$, as is expected of a spin-1 operator.

In an integrable relativistic QFT, there exists, beside the energy and momenta, an infinite number of local conserved charges, which transform as higher Lorentz spin operators. These conserved charges can be parametrized in terms of their light-cone components as

$$P_s^{\pm}|\{\theta_i\}\rangle = \sum_i p_s e^{\pm s\theta_i}|\{\theta_i\}\rangle, \tag{2}$$

where $p_s$ is some constant, and $s$ labels the spin of the conserved charge. The expression (2) takes into account the fact that these charges commute with $H$, transform under a boost as a spin-$s$ operator, and are local, since when acting on a multiparticle state, their eigenvalues can be written as a sum over separable contributions from each individual particle.

The existence of this infinite set of charges places very strong constraints on the particle scattering. We can for instance easily derive the fact that scattering is always elastic in an integrable QFT. In a scattering event, we assume we have an incoming state at $t \to -\infty$ given by $|\{\theta_i\}\rangle_{\text{in}}$, and after these particles scatter, we have an outgoing state at $t \to \infty$, $|\{\theta_i'\}\rangle_{\text{out}}$. The existence of the charges (2) implies the infinite set of conditions,

$$\sum_i p_s e^{\pm s\theta_i'} = \sum_i p_s e^{\pm s\theta_i}.$$

This infinite set of conditions can only be satisfied if the set of incoming and outgoing rapidities are exactly equal, $\{\theta_i'\} = \{\theta_i\}$. This is the statement of elastic scattering, scattering preserves the full set of particle momenta.

It is important to point out that this result relies on the fact that the function $e^{\pm s\theta}$ is positive-definite for all real values of rapidities. This is connected to the fact that there is an infinite subset of conserved charges which are parity-even (which are invariant under $p \to -p$). The parity-even charges are $P_s^{\text{even}} = \frac{1}{2}(P_s^+ + P_s^-)$, and parity-odd charges are given by $P_s^{\text{odd}} = \frac{1}{2}(P_s^+ - P_s^-)$. The parity-even and odd charges can be seen as generalizations of the total energy and momentum operators, respectively. The contribution from a given particle state to a parity-even charge is always positive, regardless of the sign of the particle rapidity. Parity-odd charges can have negative eigenvalues, depending on the sign of the rapidity.

It is important that there exist parity-even charges, because we can conclude scattering is elastic only if the eigenvalues of conserved charges are always positive. For instance, we should not expect elasticity if only parity-odd charges are conserved. This is because for any given state, we can add a pair of particles with opposite rapidities, $\theta, -\theta$, such that they do not produce any new contribution to the parity-odd charges. It would then always be possible to produce any number of such pairs of particles, thus violating elasticity. However, if positive-definite parity-even charges exist, the contributions to their eigenvalues from a particle cannot be canceled out by another particle, even if it has opposite rapidity.

The condition of factorization can be similarly derived. The essence of the argument relies on the fact that acting on a multi-particle state with the operators $U_s^{\pm} = \exp\left(-i\alpha P_s^{\pm}\right)$, leaves the S-matrix invariant, since, $P_s^{\pm}$ commutes with the Hamiltonian. If one thinks of particles in real space as partially localized wave packets, acting with the operator $U_s^{\pm}$ changes the position of the wave packet by an amount which depends on the expected value of the particle's momentum. Since in a multi-particle state, all particles can have different momenta, it is possible then to move the positions of all the particles individually, without producing any changes in the total S-matrix.

The statement of factorization is then that in an $N$-particle scattering process, one can separate the particles individually, such that the $N$-particle S-matrix can be written as a product of 2-particle S-matrices. Moreover, the different ways in which one can factorize the S-matrix must all yield the same result. The condition of factorization for 3-particle scattering leads to the famous Yang-Baxter equation, pictured in Fig. 1.

The conditions of elasticity and factorization, as well as other physical considerations, such as Unitarity and Lorentz invariance, place very strong constraints on the form of the S-matrix. Starting from these conditions, Zamolodchikov and Zamolodchikov outlined a Bootstrap program for computing 2-particle S-matrices in integrable QFT's [4].

We now briefly review the axioms derived from elasticity and factorization, from which the 2-particle S-matrix can be computed.

1. *Yang-Baxter equation*

   It follows from the property of factorization, that it is possible to break up any many-particle scattering process into a product of 2-particle S-matrices. We define the two-particle S-matrix $S(\theta_1, \theta_2) \equiv S(\theta_{12})$, where $\theta_{12} = \theta_1 - \theta_2$, as

   $$_{\text{out}}\langle \theta_2', \theta_1' | \theta_1, \theta_2 \rangle_{\text{in}} = S(\theta_{12})(2\pi)^2 \delta(\theta_1 - \theta_1')\delta(\theta_2 - \theta_2'),$$

   involving asymptotic incoming and outgoing particle states at $t \to -\infty$ and $t \to \infty$ respectively.

   The Yang-Baxter equation for relativistic field theories is a consistency condition requiring the equivalence of the different ways we could factorize a three-particle S-matrix in terms of two-particle S-matrices. There are two ways to factorize a three-particle scattering process, which are pictured in Figure 1. Requiring that these two factorizations yield the same three-particle S-matrix, we then have the relation

   $$S(\theta_{12})S(\theta_{13})S(\theta_{23}) = S(\theta_{13})S(\theta_{23})S(\theta_{12}). \tag{3}$$

   Equation (3) is trivially satisfied in systems with only one species of particle, without internal quantum number structure. The Yang-Baxter equation becomes a nontrivial constraint when the particles have some internal structure, labeled by some index, such that particle states can be written as $|\theta, a\rangle$, where $a$ runs over the internal quantum numbers. The two-particle S-matrix then involves the index structure of the two incoming and outgoing particles, and can be written as

   $$_{\text{out}}\langle \theta_2', d; \theta_1', c | \theta_1, a; \theta_2, b \rangle_{\text{in}} = S_{ab}^{dc}(\theta_{12})(2\pi)^2 \delta(\theta_1 - \theta_1')\delta(\theta_2 - \theta_2').$$

   The general Yang-Baxter equation when there are internal quantum numbers, is then

   $$S_{ab}^{hg}(\theta_{12})S_{ge}^{ic}(\theta_{13})S_{hi}^{fd}(\theta_{23}) = S_{ah}^{fi}(\theta_{13})S_{be}^{hg}(\theta_{23})S_{ig}^{dc}(\theta_{12}),$$

   where summation over repeated indices is implied.

2. *Unitarity axiom*

   This next axiom follows from the assumption that the time evolution is unitary, applied to the 2-particle elastic S-matrix. Unitarity implies that no information is lost under time evolution, meaning that a scattering process can be reversed. This results in the consistency condition pictured in Figure 2. This unitarity axiom is expressed explicitly as

   $$S(\theta)S(-\theta) = 1. \tag{4}$$

3. *Crossing symmetry*

   A feature of relativistic QFT's is that scattering amplitudes are invariant under crossing symmetry. The amplitude for a scattering process with a particle (antiparticle) in the

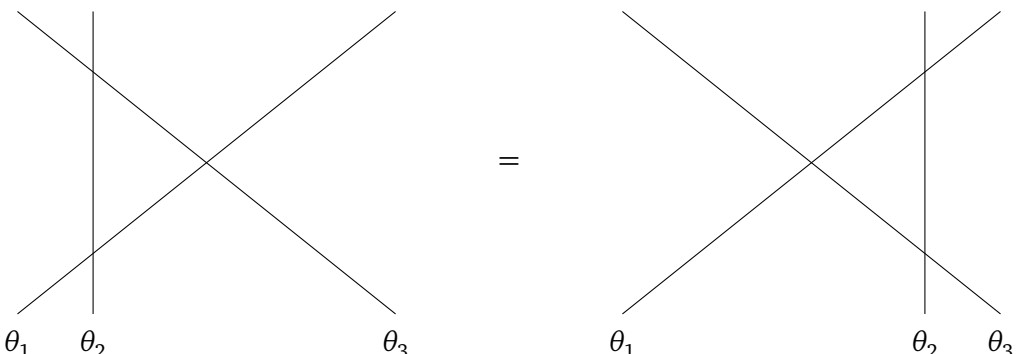

Figure 1: Yang-Baxter equation.

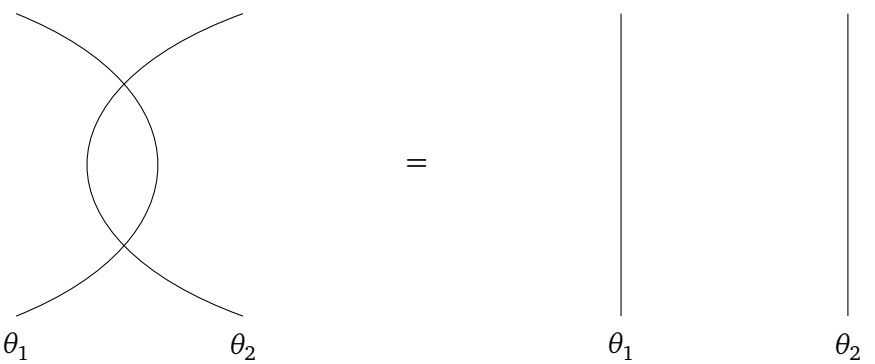

Figure 2: Unitarity axiom.

incoming state, is equivalent to the same amplitude instead with an antiparticle (particle) in the outgoing state, and vice versa, after shifting the momentum and energy of the particle as $p \rightarrow -p$, $E \rightarrow -E$. The procedure of shifting a particle from an incoming state into an antiparticle in the outgoing state, is called crossing. In terms of the rapidity, particles are crossed by shifting $\theta \rightarrow \theta + \pi i$.

If we consider a QFT which has only one type of real particle (which is its own antiparticle), then crossing symmetry leads to the consistency condition pictured in Figure 3. In terms of the S-matrix, this implies the condition

$$S(\theta) = S(\pi i - \theta).$$ (5)

In more general cases where particles and antiparticles are distinguishable, crossing symmetry leads to conditions analogous to (5) relating the particle-particle, particle-antiparticle, and antiparticle-antiparticle S-matrices.

4. *Bound state bootstrap axiom*

Under some circumstances, it is possible that the interaction between two particles lead to the formation of a bound state. In a general QFT, very massive bound states may be kinematically unstable, and decay into lighter particles. In an Integrable QFT, elasticity implies that all bound states are stable particles, and can be treated in equal footing as the elementary particles.

Bound states can appear as intermediate states in the scattering amplitude of two elementary particles. If there exist two elementary particles, labeled by some quantum

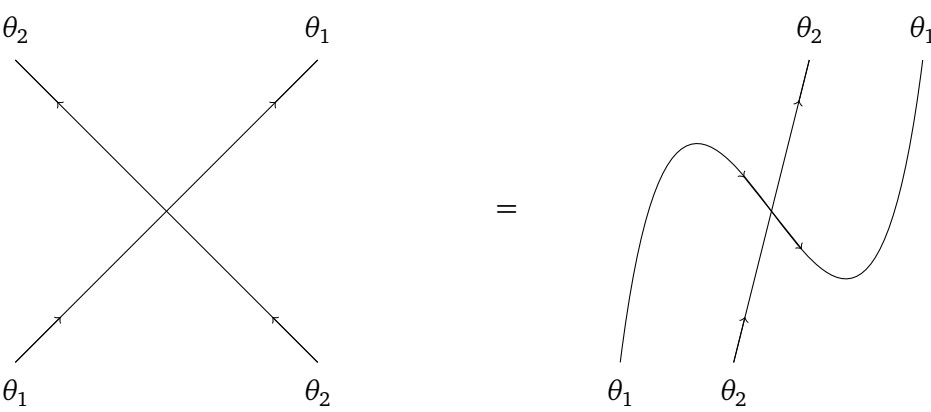

Figure 3: Crossing symmetry.

numbers, $a$ and $b$ respectively, and these two can form a bound state particle labeled by a quantum number, $c$, this is reflected in the fact that the S-matrix between the two elementary particles, $S_{ab}(\theta)$, will have a pole at some "fusion angle", $iu_{ab}^c$, such that

$$\text{Res}_{\theta \to iu_{ab}^c} S_{ab}(\theta) = i(\Gamma_{ab}^c)^2,$$

where $\Gamma_{ab}^c$ is some constant corresponding to the amplitude of the three-particle vertex, and $u_{ab}^c$ is a positive real number. The masses of the three particles are related by

$$m_c^2 = m_a^2 + m_b^2 + m_a m_b \cos u_{ab}^c.$$

Since the elementary particles and bound states can be treated on equal footing, it is possible that any two of the three particles, $a$, $b$, $c$, may fuse to form the third particle, with the consistency condition on the three possible fusion angles,

$$u_{ab}^c + u_{bc}^a + u_{ca}^b = 2\pi.$$

Our final axiom we impose on the S-matrix is the so-called bound state bootstrap axiom, which is pictured in Figure 4. This axiom involves two particles, $a$, $b$ which may form a bound state, $c$, and which are scattering with another particle, $d$. We then need a consistency condition relating the scattering of $d$ with the two elementary particles, and the scattering of $d$ with the intermediate-state bound state, which results in the axiom

$$S_{da}(\theta + i\bar{u}_{ca}^b)S_{db}(\theta - i\bar{u}_{bc}^a) = S_{dc}(\theta), \tag{6}$$

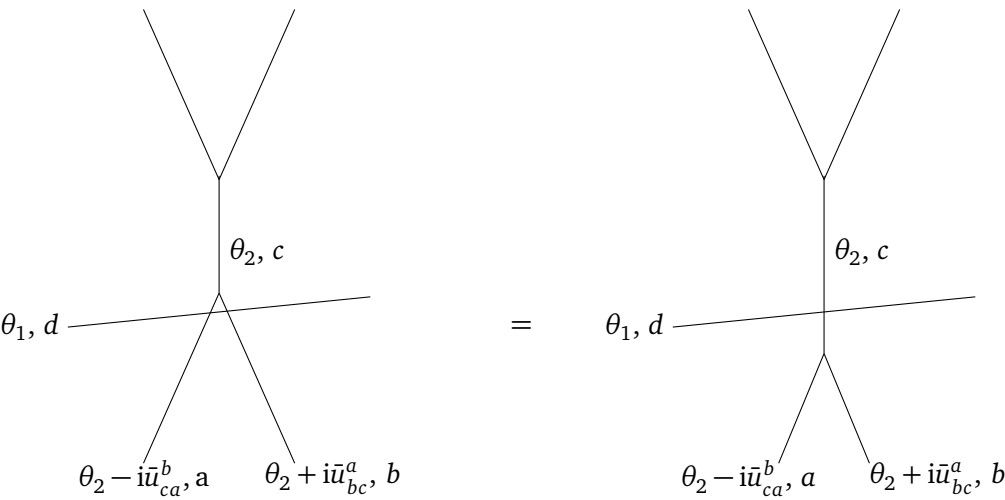

Figure 4: Bound-state bootstrap axiom.

where $\bar{u}_{ab}^c \equiv \pi - u_{ab}^c$.

# 3 Floquet Hamiltonians and integrability

We now consider the dynamics of periodically driven quantum field theories. In particular, we study a system whose time-evolution is given by the time-dependent Hamiltonian $H(t)$, such that $H(t + T) = H(t)$, for some driving period $T$. Such periodically driven systems can be understood via the Floquet theorem [29, 30], which states that the unitary time-evolution operator may be written as

$$U(t) = P(t)e^{-iH_F t}, \tag{7}$$

where $H_F$ is a time-independent Hamiltonian, usually called the "Floquet Hamiltonian", and $P(t)$ is a periodic unitary operator, such that $P(t + T) = P(t)$.

Theorem (7) implies that the stroboscopic time evolution (where one measures the system after time intervals which are integer multiples of $T$, and is not concerned with the microscopic time-evolution within a single period) is governed only by the Floquet Hamiltonian, $H_F$. It is then extremely useful to understand the properties of $H_F$ for any given protocol, such as its eigenstates and locality properties.

It is known that even in very simple driving protocols, the Floquet Hamiltonian may be qualitatively very different from conventional Hamiltonians. For instance, one can consider the two-step process, where the stroboscopic time-evolution is given by

$$U(T) = e^{-iH_F T} = e^{-iH_2 T_2}e^{-iH_1 T_1}, \tag{8}$$

where $T = T_1 + T_2$. In the simplest scenario, we can consider $H_1$ and $H_2$ to be integrable Hamiltonians, with $[H_1, H_2] \neq 0$. It is well understood that even in this simplest possible scenario the driving protocol generally leads to a non-integrable Floquet Hamiltonian. For the two-step protocol this can be easily understood through the Baker-Campbell-Hausdorff (BCH)

formula, which allows us to compute the the Hamiltonian as

$$
\begin{aligned}
H_F T &= \mathrm{i}\log\left(e^{-\mathrm{i}H_2 T_2} e^{-\mathrm{i}H_1 T_1}\right) \\
&= H_2 T_2 + H_1 T_1 - \frac{\mathrm{i}}{2}[H_2, H_1] T_2 T_1 \\
&\quad - \frac{1}{12}([H_2,[H_2,H_1]]T_2^2 T_1 + [H_1,[H_1,H_2]]T_2 T_1^2) + \dots
\end{aligned}
\tag{9}
$$

This expansion generally generates an infinite number of terms that contribute to $H_F$, which generally break any nontrivial conservation laws of $H_1$ and $H_2$, therefore breaking integrability.

Besides breaking integrability, the BCH expansion for $H_F$ reveals an even worse problem: Floquet Hamiltonians are generally highly nonlocal. This can be easily observed, for example, if $H_1$ and $H_2$ describe typical integrable quantum spin chains, such as the transverse field Ising chain, or Heisenberg chains. In this case, the two Hamiltonians involve only interactions between neighboring spins, and are thus local. The higher terms of the BCH expansion produce longer range interaction terms. For example, the third term in the right-hand-side of (9), leads to a three-spin interaction term. This behavior extends to the higher terms, the more nested commutators involved in the term, it will contribute couplings involving a higher number of spins.

A proposed consequence of the non-locality of $H_F$ is that the system generally heats up with the driving, eventually reaching a state that is indistinguishable from an infinite-temperature state [20]. This is because the eigenstates of the highly nonlocal $H_F$ may be indistinguishable from infinite-temperature states of a local Hamiltonian.

Another notable problem with the BCH expansion is that it is not generally guaranteed to converge, specially for large driving periods [20]. Some progress has been recently achieved by resumming some of the terms in the BCH expansion using a replica trick and formulating the problem as an expansion in the strength of the periodic driving [31].

Despite the complexity of Floquet Hamiltonians in general, one may ask if it is possible that there are special cases of driving protocols that preserve some form of integrability. To answer this question one needs to have a precise definition of what is meant by integrability in a periodically driven system. This question has been already raised in [19], where various possible definitions were suggested. In this paper, our definition of Floquet integrability will be very similar to the standard definition of integrability in equilibrium QFT. We define an integrable Floquet QFT protocol as one such that there is an infinite number of *independent* and *local* charges that commute with the Floquet Hamiltonian. Further, we require that an infinite subset of these charges be parity-even, or positive definite.

Our condition for integrability is stronger than that which applies to some of the examples presented in [19], where integrability was defined as systems that do not "heat up" under periodic driving. For this requirement to be fulfilled, it is sufficient (though not necessary) to show that the Floquet Hamiltonian is local, which implies traditional energy conservation. As we will later discuss, it is also possible to have driving protocols where the total energy of the system increases indefinitely with each cycle, yet the system does not "heat up", in the sense that the entropy doesn't increase. In contrast, our definition of integrability is much stronger in that we do not require that only energy or entropy are conserved, but we require an infinite number of other quantities to be conserved as well.

We also point out that driving protocols have been studied, where late-time dynamics can be described by a time-periodic version of the generalized Gibbs ensemble (pGGE) [32, 33]. This implies that in such cases an infinite number of quantities are conserved under stroboscopic time evolution. The existence of a pGGE, however, does not necessarily guarantee that the system is integrable, and solvable in the sense defined in this paper. This is because

the infinite conserved charges involved in the pGGE may be highly nonlocal, in the same way that Floquet Hamiltonians are also nonlocal.

We will restrict ourselves to look for driving protocols where for some parts of the period, $T$, the Hamiltonian is that of an integrable QFT, such that the time-dependent Hamiltonian can be written as

$$H(t) = \begin{cases} H'(t), & \text{for } t \in (0, T_1], \text{ mod } T \\ H_{\text{int}}, & \text{for } t \in (T_1, T], \text{ mod } T, \end{cases} \tag{10}$$

where $H_{\text{int}}$ is the Hamiltonian of some integrable QFT, and $H'(t)$ is some time dependent Hamiltonian. We assume that it is possible that some such protocol (10) exists, such that the corresponding Floquet Hamiltonian is integrable. We then analyse the physical consequences of this Floquet integrability. For simplicity, in most of our analysis we will consider $H_{\text{int}}$ to describe a field theory whose spectrum consists of only one species of particle. It is not difficult to generalize our results to other situations.

We now consider the following protocol: for $t < 0$, the system is prepared to be in some eigenstate of $H_{\text{int}}$, then for $t \geq 0$, we start driving with the Hamiltonian (10). The state of the system at time $t = 0 - \epsilon$ (for very small real and positive, $\epsilon \to 0$), can be written as some multi-particle state given by

$$|\Psi(0 - \epsilon)\rangle = |\{\theta_i\}\rangle_{\text{in}}. \tag{11}$$

We can now stroboscopically evolve this initial state with the integrable Hamiltonian, $H_F$, to obtain the state at time $nT - \epsilon$, after $n$ full periods. The state at this time can also generally be written as a superposition of the eigenstates of $H_{\text{int}}$, such that

$$|\Psi(nT - \epsilon)\rangle = e^{-iH_F nT}|\Psi(0 - \epsilon)\rangle = \sum_a C_a|\{\theta_i'\}_a\rangle, \tag{12}$$

where the label, $a$, denotes a sum over different sets of configurations of particle rapidities. This representation is always possible, given that the basis of particle eigenstates of $H_{\text{int}}$ is complete and spans the Hilbert space. Unitarity of the time evolution requires that the coefficients be normalized as $\sum_a |C_a|^2 = 1$.

We now examine what restriction does integrability of $H_F$ place on the form of the $n$-period states (12). We assume there exist an infinite number of *independent* and *local* charges $Q_s$, labeled by some parameter, $s$, which commute with $H_F$. We further demand that some infinite subset of these charges be parity-even.

If we consider a large enough number of periods, $n$, locality of the conserved charges, $Q_s$ implies that their expectation values on the final states (12), are given by the sum of individual contributions from each particle. This assumption that the final state can be represented in terms of asymptotic particle states is sensible in translationally invariant systems; it is not necessarily satisfied in spatially confined systems, such as the quantum Newton's cradle [34], where the Hilbert space may not be spanned by asymptotic states with well separated particles.

Locality of the conserved charges, means that if we act with $Q_s$ on an asymptotic particle state, the result can be written as a sum of individual contributions from each particle in the state. Acting on a given eigenstate with well separated particles, we then expect

$$\langle\{\theta_i\}|Q_s|\{\theta_i\}\rangle = \sum_i q_s(\theta_i),$$

for some function $q_s(\theta)$. Considering the parity-even conserved charges, we can be sure that there is an infinite subset of charges for which the function $q_s(\theta)$ is positive-definite for all

real values of $\theta$. The fact that $Q_s$ commutes with $H_F$ implies that its expectation value on the states (11) and (12) must be equal, such that

$$\langle\Psi(0-\epsilon)|Q_s|\Psi(0-\epsilon)\rangle = \langle\Psi(nT-\epsilon)|Q_s|\Psi(nT-\epsilon)\rangle,$$

or explicitly

$$\sum_i q_s(\theta_i) = \sum_a \sum_i |C_a|^2 q_s(\theta'_{i,a}). \tag{13}$$

The assumption of *independence* of the charges $Q_s$ implies that $q_s(\theta)$ form an infinite set of independent functions, for different values of $s$. We can then apply the same reasoning from equilibrium integrable QFT, that the only way to always satisfy the infinite set of independent conditions (13), is if the final state consists of the same set of rapidities as the initial state. The final state (12) can then be written as a single eigenstate, eliminating the sum over $a$, and with $\{\theta_i\} = \{\theta'_i\}$.

We have then shown that if an integrable Floquet protocol exists, this means that the stroboscopic time evolution is elastic, *i.e.* the set of particle momenta is conserved. This elastic property, however, does not necessarily apply to the microscopic time evolution within single periods. It might be possible that for some intermediate time, different sets of particles are created, or the momenta of existing particles may be shifted, but once the full period is completed, the particle content needs to be restored. We will see some explicit examples of this stroboscopic elasticity in later sections.

We have established that stroboscopic time evolution is elastic in an integrable Floquet protocol, by using similar arguments as those from equilibrium integrable QFT. It is also similarly easy to show that the stroboscopic time evolution in these systems is also factorizable. The argument is the same as the equilibrium argument discussed in the previous section. Suppose we act on a given multi-particle state with the operators $U_s = \exp(-i\alpha Q_s)$, which commute with the stroboscopic time-evolution operator, $\exp(-iH_F nT)$. The operators $U_s$ shift the expected position of a particle by an amount which depends on the particle momentum and on the particular index $s$, so all the particle positions are shifted by different amounts. The particle trajectories can then be independently shifted while leaving the stroboscopic time evolution invariant, which is the statement of factorization.

The properties of stroboscopic elasticity and factorization are illustrated in Fig. 5.

As we discussed in the previous section, the properties of elasticity and factorization can be used in equilibrium integrable QFT to establish a bootstrap program, through which one can compute exact S-matrices. In the next section we will show how the properties we have presented, of stroboscopic elasticity and factorization can be used to establish an analogue version of a bootstrap approach to compute exactly the effect of driving in the time evolution of an integrable Floquet QFT.

## 4 Integrable Floquet bootstrap program

We will focus on driving protocols of the form (10), where for some portion of the period the Hamiltonian is that of an integrable QFT, $H_{\text{int}}$. For simplicity of presentation, we focus on QFT's with only one type of particle, whose S-matrix is known, and denoted by $S(\theta)$.

We now examine the situation where an initial state is given by a one-particle eigenstate, $|\theta\rangle$, of $H_{\text{int}}$. In the absence of driving, if we were simply evolving with $H_{\text{int}}$, the state of the system at time $T$ would be given by

$$|\theta\rangle_T = e^{-iTm\cosh\theta}|\theta\rangle. \qquad \text{(no driving)}$$

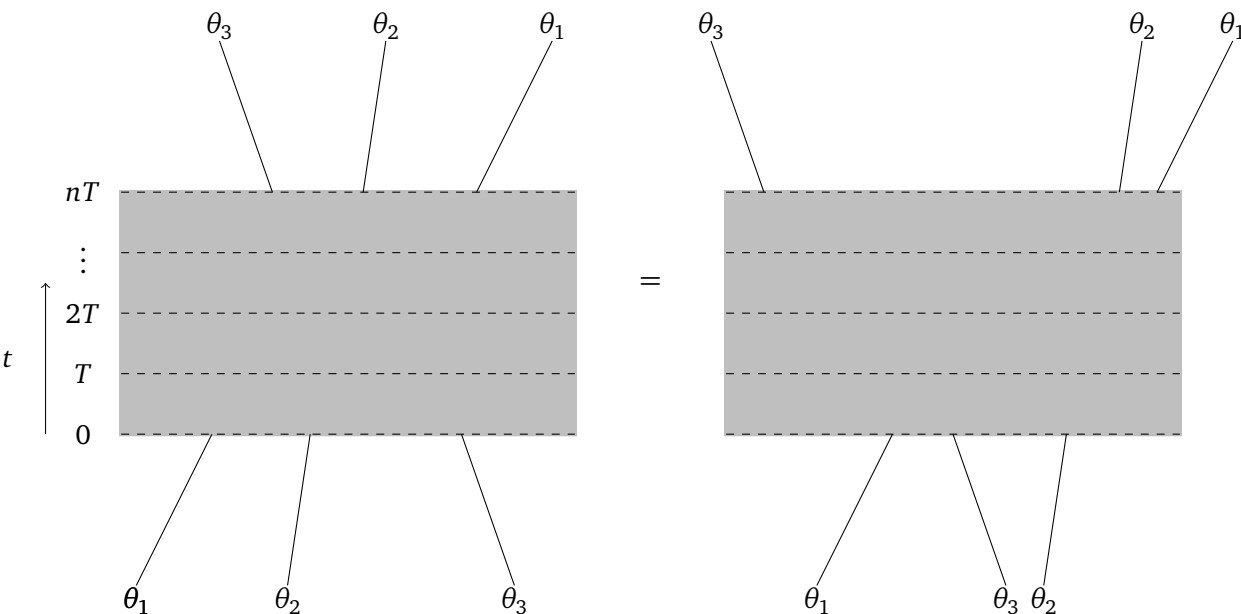

Figure 5: We picture schematically the stroboscopic evolution of particle trajectories with rapidities $\theta_{1,2,3}$, after $n$ full periods, under an integrable Floquet Hamiltonian. The property of *elasticity* is displayed in the fact that after $n$ full periods, the particle content and set of rapidities is conserved. Note that there does not need to be elasticity at microscopic times within a single period. The property of *factorization* is displayed by the fact that the positions of particle trajectories can be shifted individually, but the amplitude for the two depicted processes must be equivalent.

If we turn on the driving, but the Floquet Hamiltonian is integrable, from the property of stroboscopic elasticity, we expect that after n full periods, $nT$, the system is again in a one-particle state. In general, we can then write the state at time $nT$ as

$$|\theta\rangle_{nT} = e^{-inTm\cosh\theta} F^n(\theta)|\theta\rangle, \tag{14}$$

where $F^n(\theta)$ captures all the effects of periodic driving. We can interpret $F(\theta)$ as an additional phase acquired by the particle after each driving period.

We now assume the simplest way to satisfy the time evolution (14), is that the particle content is conserved, stroboscopically, after every period, $T$, such that

$$|\theta\rangle_T = e^{-iTm\cosh\theta} F(\theta)|\theta\rangle, \tag{15}$$

From the property of factorization, we expect that time evolution (15) can be also applied to multi-particle states. This is because we can shift the positions of particle trajectories arbitrarily. We can then widely separate all the particles, so we expect the effect of driving on the stroboscopic time evolution is factorizable, and can be expressed in terms of the phase, $F(\theta)$ of individual particles. Then we expect that after a full period, $T$, the time evolution of a multi-particle state, with ordered rapidities, $\theta_1 < \theta_2 < \dots \theta_k$, can be written as

$$
\begin{aligned}
|\theta_1,\theta_2,\dots,\theta_k\rangle_T &= e^{-iTm\sum_{i=1}^{k}\cosh\theta_i} F(\theta_1,\theta_2\dots,\theta_k)|\theta_1,\theta_2,\dots,\theta_k\rangle \\
&= e^{-iTm\sum_{i=1}^{k}\cosh\theta_i}\left(\prod_{i=1}^{k}F(\theta_i)\right)|\theta_1,\theta_2,\dots,\theta_k\rangle. \tag{16}
\end{aligned}
$$

We have then reduced the problem of computing the stroboscopic time evolution of a given state, to that of finding the function $F(\theta)$. We remark that this is very similar to what is done

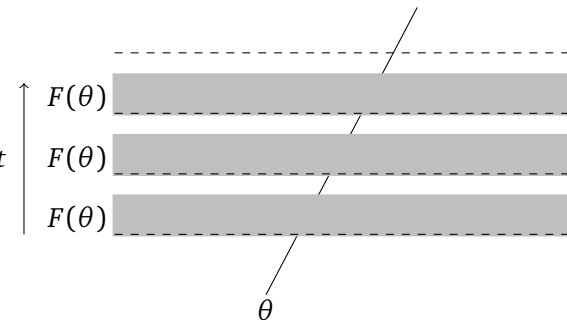

Figure 6: A particle propagating under an integrable Floquet Hamiltonian corresponding to the protocol Eq. (10). When measured stroboscopically, the particle rapidity is conserved. The effect of driving is only that the particle time evolution acquires a factor of $F(\theta)$ with every period, here represented as crossing through shaded region. The exact particle content and details of the dynamics within the shaded region are unknown.

in equilibrium integrable QFT, where the multi-particle scattering problem is reduced to just finding the function $S(\theta)$.

Taking inspiration from the S-matrix bootstrap program, we now list a set of axioms that will strongly constrain the function $F(\theta)$. Similarly to the S-matrix bootstrap program, the axioms that follow are consequences of elasticity, factorization, unitarity and Lorentz invariance:

1. *Floquet Yang-Baxter equation:*

   If the stroboscopic time evolution is integrable, and of the form (10), the function $F(\theta)$ must satisfy what we call the *Floquet Yang-Baxter equation*, which is depicted in Fig. 7. We consider an initial state with two particles with rapidities, $\theta_1$, $\theta_2$. Using factorization, we can shift the particle trajectories such that the two particle collide at some time $t \in (T_1, T) \bmod T$ when the time evolution is given by $H_{\text{int}}$. It follows from factorization that we can shift the particle trajectories such that the collision happens before or after a period $T$, and both processes must be equivalent.

   We note that using the stroboscopic factorization properties, we can also shift the particle trajectories, such that the collision between the two particles would happen somewhere in the shaded regions governed by $H'(t)$. This would correspond to a combined phase in the time evolution, $F(\theta_1, \theta_2)$. Stroboscopic factorization demands that whether the collision happens before, during or after a given shaded region, the amplitude of the physical process must be equivalent. This leads to the equation

$$F(\theta_1, \theta_2) = F(\theta_1)F(\theta_2)S(\theta_1 - \theta_2) = S(\theta_1 - \theta_2)F(\theta_2)F(\theta_1). \tag{17}$$

   This equation then fixes the function $F(\theta_1, \theta_2)$ in terms of the simpler functions $F(\theta)$, $S(\theta$. In the case where the spectrum of $H_{\text{int}}$ consists of only one species of particle, the second equality in Eq. (17) is trivially satisfied. This condition is, however, much more important for field theories with many types of particles, and for which the S-matrix is not diagonal, for which we can write matrix-valued version of (17):

$$F_a^e(\theta_1)F_b^f(\theta_2)S_{ef}^{cd}(\theta_1 - \theta_2) = S_{ab}^{ef}(\theta_1 - \theta_2)F_f^d(\theta_2)F_e^c(\theta_1),$$

   where the indices run over the internal quantum numbers of incoming and outgoing particles.

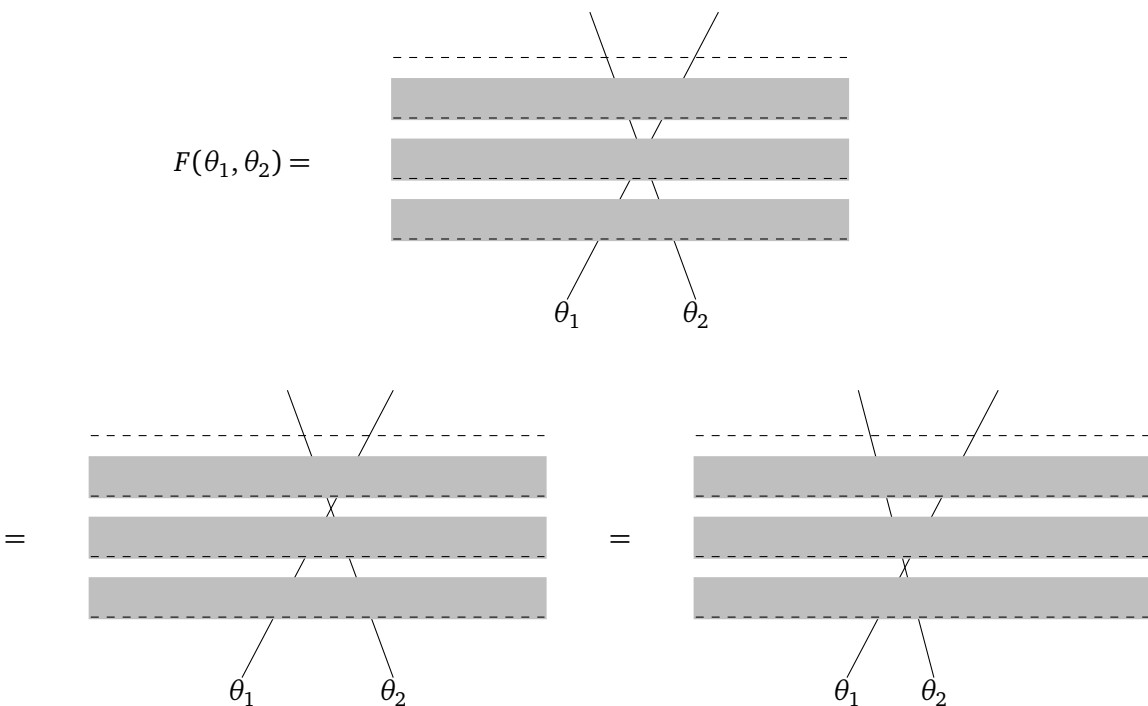

Figure 7: Floquet Yang-Baxter equation.

2. *Floquet annihilation axiom*

   In a relativistic field theory, there exists the possibility that a particle may annihilate with its antiparticle. As a consequence, it is known that form factors of local operators possess singularities at configurations of rapidities where an incoming particle and antiparticle annihilate [6]. If the initial state contains a particle with rapidity $\theta$, an "annihilation pole" exists in form factors of local operators if there is also an incoming antiparticle with rapidity $\theta + i\pi$. If we demand that Floquet integrability is consistent with annihilation of particles and antiparticles, we arrive at the constraint depicted in Fig. 8, which we call the *Floquet annihilation axiom*. Explicitly, the condition on the function $F(\theta)$ that ensures compatibility with annihilation is

$$\boxed{F(\theta)F(\theta + i\pi) = 1.} \tag{18}$$

   It will be later useful to rewrite Eq. (18) in terms of the particle energy and momentum, instead of the rapidity, such that

$$F(E, p)F(-E, -p) = 1.$$

3. *Floquet cross-unitarity axiom/Parity invariance*

   This next axiom follows from demanding compatibility between Floquet integrability, S-matrix unitarity and relativistic invariance. We start by performing a space-time rotation, where we exchange the role of the spatial and temporal directions. In terms of particle content, this rotation is done by shifting all the rapidities as $\theta \rightarrow i\pi/2 - \theta$. In this picture, instead of having a periodically driven system, we consider a system with spatial defects which are periodic in $x$. In this space-time rotated theory, Floquet integrability translates into the fact that scattering of a particle against one of the defects is elastic and

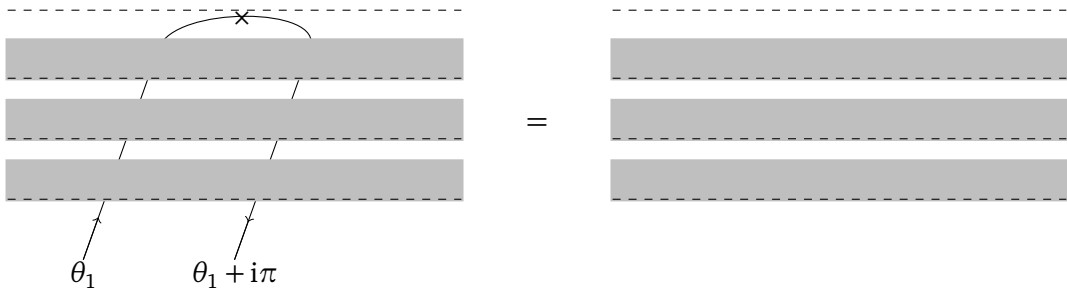

Figure 8: Floquet annihilation axiom.

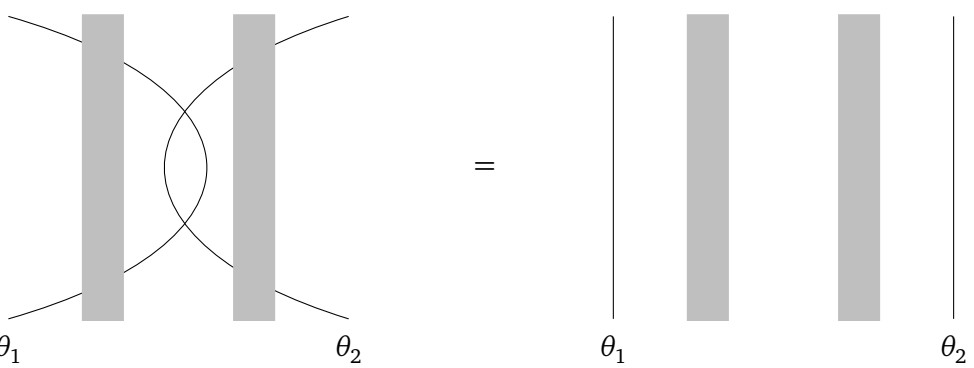

Figure 9: Floquet cross-unitarity axiom.

factorizable. Applying the property of S-matrix unitarity in this rotated channel leads us to our next axiom, pictured in Fig. 9. We call this axiom the *Floquet cross-unitarity axiom*, and the explicit condition on $F(\theta)$ is given by

$$F\left(\frac{i\pi}{2} - \theta\right) F\left(\frac{i\pi}{2} + \theta\right) = 1. \tag{19}$$

We can combine the condition (19) with the annihilation axiom (18) to obtain the condition

$$F(\theta) = F(-\theta), \tag{20}$$

which simply implies that the function $F(\theta)$ is symmetric under a parity transformation. Only two of the three conditions, (18), (19) and (20) are independent constraints on $F(\theta)$.

4. *Floquet bound-state bootstrap axiom*

This next axiom applies for integrable QFT's whose spectrum contains excitations which are bound states of other particles. As we discussed in Section 2, if there are particles with quantum numbers labeled by $a, b, c$, and the $c$ particle can be considered to be a bound state of an $a$ and $b$ particle, then there is a fusion angle $u_{ab}^c$, such that the S-matrix, $S_{ab}(\theta)$ has a pole at $\theta = i u_{ab}^c$. Using the property of stroboscopic factorization, we can shift the point in time when two particles fuse to form a bound state. This leads to the condition pictured in Fig. 10, which we call the *Floquet bound-state bootstrap*

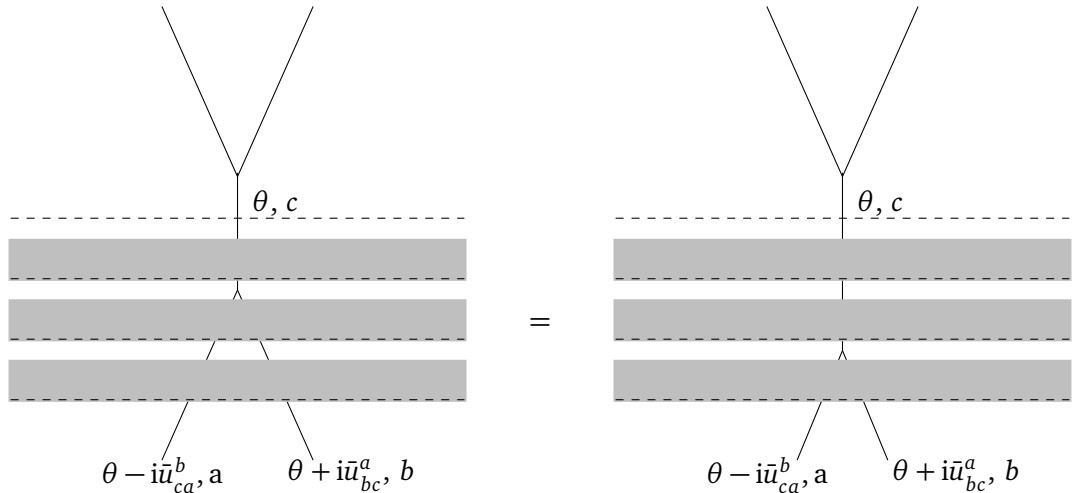

Figure 10: Floquet bound-state bootstrap axiom.

*axiom*, or explicitly,

$$F_a(\theta + i\bar{u}^b_{ca})F_b\left(\theta - i\bar{u}^a_{bc}\right) = F_c(\theta),$$
(21)

where $F_{a,b,c}(\theta)$ is the stroboscopic evolution function corresponding to each type of particle.

We point out that one simple function that satisfies all the constraints on $F(\theta)$ is the standard particle time evolution phase, $F(\theta) = \exp(-iT'm\cosh\theta)$, for some duration $T'$.

Another simple solution is given by a constant, $F(\theta) = \pm 1$. The solution with the positive sign corresponds to the trivial scenario where there is no driving, such that $H'(t) = H_{\text{int}}$, in the protocol (10).

We finally point out a very useful aspect of having integrable Floquet dynamics, which is that, once the function $F(\theta)$ has been identified, it is possible to compute correlation functions of stroboscopically-separated operators. For example, we look at the two point function of operators $\mathcal{O}_1(x, 0)$ and $\mathcal{O}_2(y, nT)$. Assuming the initial state is the ground state of $H_{\text{int}}$, the two-point function can be written as

$$C_{2,1}(y, nT; x, 0) \equiv \langle 0|\mathcal{O}_2(y, nT)\mathcal{O}_1(x, 0)|0\rangle.$$
(22)

The two-point function (22) can be computed by inserting a sum over the complete set of states between the two operators, and applying the known properties of the stroboscopic time evolution of particles. We then express (22) as

$$
\begin{aligned}
C_{2,1}(y, nT; x, 0) &= \sum_{k=0}^{\infty} \int \frac{d\theta_1 \dots d\theta_k}{k!(2\pi)^k} \langle 0|\mathcal{O}_2(0,0)|\theta_1, \dots \theta_k\rangle \left[\langle 0|\mathcal{O}_1(0,0)|\theta_1, \dots \theta_k\rangle\right]^* \\
&\times \exp\left(-im(y-x)\sum_{i=1}^{k}\sinh\theta_i - imnT\sum_{i=1}^{k}\cosh\theta_i\right)\prod_{i=1}^{k}[F(\theta_i)]^n.
\end{aligned}
$$
(23)

It is then clear that if the matrix elements of operators between particle states are known (which may be found using the form factor bootstrap program [6]), and the stroboscopic time-evolution function, $F(\theta)$, is known, then it is possible to compute correlation functions of operators which are separated by stroboscopic time scales.

In the next section we will study a simple two-step driving protocol for free bosonic and fermionic QFT's. These protocols do not generally yield an integrable Floquet Hamiltonian, however, it will be instructive to see precisely how integrability is broken. This understanding will make it easier to identify how integrability can arise in some other driving protocols.

In Section 6, we will then present several simple examples where the Floquet Hamiltonian is integrable, and we can compute explicitly the corresponding functions, $F(\theta)$, which satisfy all the axioms that have been proposed in this section.

## 5   Floquet dynamics of free field theories: an instructive counter-example

In this section we consider a simple two-step driving protocol, first for a free bosonic field, and then for a free fermion. For both cases, the driving protocol will consist of alternating between two different values of the particle mass, $m_1$ and $m_2$. The stroboscopic time-evolution operator is given by

$$U(T) = e^{-iT_1 H_{m_1}} e^{-iT_2 H_{m_2}},$$

where $H_m$ is the Hamiltonian corresponding to a free boson, or fermion of mass $m$. Additional constraints arise from demanding that the physical fields be continuous as functions of time after every driving step.

### 5.1   Free boson

We now consider a free bosonic field, $\phi$, with mass $m_1$ and Lagrangian,

$$\mathcal{L} = \frac{1}{2}(\partial_t \phi)^2 - \frac{1}{2}(\partial_x \phi)^2 - \frac{1}{2}m_1^2 \phi^2.$$

The field can be parametrized in terms of particle creation and annihilation operators,

$$\phi(x) = \int \frac{dp}{2\pi} \frac{1}{\sqrt{2E_{1,p}}} e^{ipx} \left( a_{1,p} + a_{1,-p}^\dagger \right), \tag{24}$$

where $a_{1,p}^\dagger$ creates, a particle with momentum, $p$, with dispersion relation given by

$$E_{1,p}^2 = m_1^2 + p^2.$$

We can also write the conjugate momentum field as

$$\Pi(x) = \int \frac{dp}{2\pi}(-i)e^{ipx} \sqrt{\frac{E_{1,p}}{2}} \left( a_{1,p} - a_{1,-p}^\dagger \right).$$

The corresponding Hamiltonian can be written as

$$H_{m_1} = \int \frac{dp}{2\pi} E_{1,p} a_{1,p}^\dagger a_{1,p}.$$

We now consider the following protocol. At times $t < 0$, the system is prepared in some eigenstate, $|\Psi(0)\rangle$ of the Hamiltonian $H_{m_1}$. At $t = 0$ we then start driving the system with the Hamiltonian $H_{m_2}$, corresponding to a free boson with mass $m_2 \neq m_1$. At time $t = T_2$, we switch back to the Hamiltonian $H_1$, until time $T_1 + T_2$, and keep driving periodically this way.

We demand that at every step where we switch the Hamiltonian, the fields $\phi(x)$ and $\Pi(x)$ are continuous.

Using the continuity of the physical fields, we can find at any given time, relations between the creation and annihilation operators at different driving steps. For example, demanding that the fields be continuous at $t = 0$, leads to the conditions

$$
\frac{1}{\sqrt{2E_{1,p}}}\left(a_{1,p} + a_{1,-p}^{\dagger}\right) = \frac{1}{\sqrt{2E_{2,p}}}\left(a_{2,p} + a_{2,-p}^{\dagger}\right),
$$

$$
\sqrt{\frac{E_{1,p}}{2}}\left(a_{1,p} - a_{1,-p}^{\dagger}\right) = \sqrt{\frac{E_{2,p}}{2}}\left(a_{2,p} - a_{2,-p}^{\dagger}\right). \tag{25}
$$

The creation and annihilation operators accross the $t = 0$ boundary are then related to each other by a Bogoliubov transformation,

$$
\begin{aligned}
a_{2,p} &= c_{(1),p}\, a_{1,p} + d_{(1),p}\, a_{1,-p}^{\dagger}, \\
a_{2,-p}^{\dagger} &= c_{(1),p}\, a_{1,-p}^{\dagger} + d_{(1),p}\, a_{1,p},
\end{aligned} \tag{26}
$$

with coefficients,

$$
c_{(1),p} = \frac{1}{2}\left(\sqrt{\frac{E_{2,p}}{E_{1,p}}} + \sqrt{\frac{E_{1,p}}{E_{2,p}}}\right), \quad d_{(1),p} = \frac{1}{2}\left(\sqrt{\frac{E_{2,p}}{E_{1,p}}} - \sqrt{\frac{E_{1,p}}{E_{2,p}}}\right).
$$

The subscript index, $_{(1)}$, here represents that these are the coefficients corresponding to the first time we switch between the two Hamiltonians. When we switch back to the Hamiltonian $H_{m_1}$ at $t = T_2$, the corresponding Bogoliubov coefficients will have the index, $_{(2)}$, and so on.

Using the transformation (26), we are able to compute expectation values of physical observables during times $0 < t < T_2$. For example, correlation functions of the field $\phi(x)$ can be computed by writing the field as

$$
\phi(x, t = 0) = \int \frac{dp}{\sqrt{2E_{2,p}}} e^{ipx}\left(a_{2,p} + a_{2,-p}^{\dagger}\right) \tag{27}
$$

$$
= \int \frac{dp}{\sqrt{2E_{2,p}}} e^{ipx}\left[\left(c_{(1),p} + d_{(1),p}\right) a_{1,p} + \left(c_{(1),p} + d_{(1),p}\right) a_{1,-p}^{\dagger}\right]. \tag{28}
$$

This field can be evolved for times $0 < t < T_2$ by simply evolving the creation and annihilation operators in (27) as

$$
a_{2,p}(t) = e^{-iE_{2,p}t} a_{2,p}, \quad a_{2,-p}^{\dagger}(t) = e^{iE_{2,p}t} a_{2,-p}^{\dagger}. \tag{29}
$$

In terms of the expression in Eq. (28), the time evolution (29) amounts to making the Bogoliubov coefficients time dependent, and thus complex-valued, as

$$
c_{(1),p}(t) = e^{-iE_{2,p}t} c_{(1),p}, \quad d_{(1),p}(t) = e^{-iE_{2,p}t} d_{(1),p},
$$

such that

$$
\phi(x, t) = \int \frac{dp}{\sqrt{2E_{2,p}}} e^{ipx}\left[\left(c_{(1),p}(t) + d_{(1),p}^{*}(t)\right) a_{1,p} + \left(c_{(1),p}^{*}(t) + d_{(1),p(t)}\right) a_{1,-p}^{\dagger}\right]. \tag{30}
$$

Since we know how the operators $a_{1,p}$ and $a_{1,-p}^{\dagger}$ act on the initial state, the expression (30) allows us to compute any correlation functions of the field in the interval $0 < t < T_2$.

The computation we have shown until now is not new, since it corresponds to simply performing a quantum quench at $t = 0$, which has been done previously for free bosons [35]. After reaching a time greater than $T_2$, however, we switch back to the original Hamiltonian, thus departing from the standard quench dynamics.

We now demand continuity of the fields at $T_2$. At $t = T_2 - \epsilon$, the bosonic field, and its momentum conjugate are given by

$$
\begin{aligned}
\phi(x, T_2^-) &= \int \frac{dp}{\sqrt{2E_{2,p}}} e^{ipx} \left[ \left( c_{(1),p}(T_2) + d^*_{(1),p}(T_2) \right) a_{1,p} + \left( c^*_{(1),p}(T_2) + d_{(1),p}(T_2) \right) a^\dagger_{1,-p} \right], \\
\Pi(x, T_2^-) &= \int \frac{dp}{2\pi} (-i) \sqrt{\frac{E_{2,p}}{2}} \\
&\quad \left[ \left( c_{(1),p}(T_2) - d^*_{(1),p}(T_2) \right) a_{1,p} + \left( -c^*_{(1),p}(T_2) + d_{(1),p}(T_2) \right) a^\dagger_{1,-p} \right].
\end{aligned}
\tag{31}
$$

The fields at $t = T_2 + \epsilon$ can also be expressed in terms of the original creation and annihilation operators by means of a new Bogoliubov transformation, with new coefficients, $c_{(2),p}(t)$ and $d_{(2),p}(t)$, which are time-evolved as

$$
c_{(2),p}(t) = e^{-iE_{1,p}t} c_{(2),p}, \quad d_{(2),p}(t) = e^{-iE_{1,p}t} d_{(2),p}.
\tag{32}
$$

One can write a Bogoliubov transformation between the creation and annihilation operators corresponding to the field at $T_2 < t < T$ and those of the original field, as

$$
a^{(2)}_{1,p}(t) = c_{(2),p}(t) a_{1,p} + d_{(2),p}(t) a^\dagger_{1,-p}, \quad a^{\dagger(2)}_{1,-p}(t) = c^*_{(2),p}(t) a^\dagger_{1,-p} + d^*_{(2),p}(t) a_{1,p}.
\tag{33}
$$

The fields at $T_2 + \epsilon$ are then

$$
\begin{aligned}
\phi(x, T_2^+) &= \int \frac{dp}{\sqrt{2E_{1,p}}} e^{ipx} \left[ \left( c_{(2),p}(T_2) + d^*_{(2),p}(T_2) \right) a_{1,p} + \left( c^*_{(2),p}(T_2) + d_{(2),p}(T_2) \right) a^\dagger_{1,-p} \right], \\
\Pi(x, T_2^+) &= \int \frac{dp}{2\pi} (-i) \sqrt{\frac{E_{2,p}}{2}} \\
&\quad \left[ \left( c_{(1),p}(T_2) - d^*_{(1),p}(T_2) \right) a_{1,p} + \left( -c^*_{(1),p}(T_2) + d_{(1),p}(T_2) \right) a^\dagger_{1,-p} \right].
\end{aligned}
\tag{34}
$$

Demanding continuity of the fields, $\phi(x, T_2^-) = \phi(x, T_2^+)$ and $\Pi(x, T_2^-) = \Pi(x, T_2^+)$, we find the relation between Bogoliubov coefficients,

$$
\begin{aligned}
c_{(2),p} &= \frac{1}{2} e^{i(E_{1,p} - E_{2,p})T_2} \left[ \sqrt{\frac{E_{1,p}}{E_{2,p}}} + \sqrt{\frac{E_{2,p}}{E_{1,p}}} \right] c_{(1),p} \\
&\quad + \frac{1}{2} e^{i(E_{1,p} + E_{2,p})T_2} \left[ \sqrt{\frac{E_{1,p}}{E_{2,p}}} - \sqrt{\frac{E_{2,p}}{E_{1,p}}} \right] d^*_{(1),p}, \\
d^*_{(2),p} &= \frac{1}{2} e^{i(E_{1,p} - E_{2,p})T_2} \left[ \sqrt{\frac{E_{1,p}}{E_{2,p}}} - \sqrt{\frac{E_{2,p}}{E_{1,p}}} \right] c_{(1),p} \\
&\quad + \frac{1}{2} e^{i(E_{1,p} + E_{2,p})T_2} \left[ \sqrt{\frac{E_{1,p}}{E_{2,p}}} + \sqrt{\frac{E_{2,p}}{E_{1,p}}} \right] d^*_{(1),p}
\end{aligned}
\tag{35}
$$

Using the transformation (35), we are now able to compute all correlation functions of the physical fields in the time interval $T_2 < t < T$, by writing the fields at this interval in terms of the original creation and annihilation operators, as (34), and time-evolving the Bogoliubov coefficients with (32).

We can repeat a similar procedure every time we switch between Hamiltonians; by demanding continuity of the fields, we obtain new Bogoliubov coefficients. This way one can write a recursive relation between the coefficients at a given step and those from the previous step, generalizing (35) as

$$
c_{(n+1),p} = \frac{1}{2} e^{\mathrm{i}T_n(E_{[n+1]_2,p} - E_{[n]_2,p})} \left[ \sqrt{\frac{E_{[n+1]_2,p}}{E_{[n]_2,p}}} + \sqrt{\frac{E_{[n]_2,p}}{E_{[n+1]_2,p}}} \right] c_{(n),p}
$$
$$
+ \frac{1}{2} e^{\mathrm{i}T_n(E_{[n+1]_2,p} + E_{[n]_2,p})} \left[ \sqrt{\frac{E_{[n+1]_2,p}}{E_{[n]_2,p}}} - \sqrt{\frac{E_{[n]_2,p}}{E_{[n+1]_2,p}}} \right] d^*_{(n),p},
$$

$$
d^*_{(n+1),p} = \frac{1}{2} e^{\mathrm{i}T_n(E_{[n+1]_2,p} - E_{[n]_2,p})} \left[ \sqrt{\frac{E_{[n+1]_2,p}}{E_{[n]_2,p}}} - \sqrt{\frac{E_{[n]_2,p}}{E_{[n+1]_2,p}}} \right] c_{(n),p}
$$
$$
+ \frac{1}{2} e^{\mathrm{i}T_n(E_{[n+1]_2,p} + E_{[n]_2,p})} \left[ \sqrt{\frac{E_{[n+1]_2,p}}{E_{[n]_2,p}}} + \sqrt{\frac{E_{[n]_2,p}}{E_{[n+1]_2,p}}} \right] d^*_{(n),p}, \tag{36}
$$

where we have introduced the notation $[n+1]_2 = n \bmod 2$, and

$$
T_n = \begin{cases} \frac{n}{2} T, & \text{for } [n+1]_2 = 2, \\[2mm] \left(\frac{n-1}{2}\right) T + T_2, & \text{for } [n+1]_2 = 1. \end{cases}
$$

With the relations (36), it is now possible to compute correlation functions of fields at any time.

We now address the question which is the main concern of this paper. Namely, is the Floquet dynamics of a free boson we have just described integrable? If not, can we learn something about how integrability is broken?

As we have discussed in previous sections, a consequence of Floquet integrability is that the stroboscopic evolution of particles needs to be elastic. Suppose for example, that we choose the initial state to be a $k$-particle eigenstate of $H_{m_1}$ with momenta $p_1, \ldots, p_k$, namely,

$$
|\Psi(0)\rangle = |p_1, \ldots, p_k\rangle.
$$

If the Floquet Hamiltonian was integrable, we expect that at times $nT - \epsilon$, for some integer $n$, the system is again in a $k$-particle eigenstate with momenta $p_1, \ldots, p_k$, which might have acquired an additional phase, $F(p_1, \ldots, p_k)$. This condition of stroboscopic elasticity in this case is then equivalent to the requirement that

$$
d_{(2n),p} = 0, \tag{37}
$$

for every $p$, and integer $n$. In such a case, the Bogoliubov transformation between the operators at time $nT - \epsilon$, and those at $t = -\epsilon$ is

$$
a^{(2n)}_{1,p}(t) = c_{(2n),p}(t) a_{1,p}, \qquad a^{\dagger(2n)}_{1,p}(t) = c^*_{(2n),p}(t) a^\dagger_{1,p}. \tag{38}
$$

If the requirement (37) were satisfied, this would imply that $c_{(2n),p}(T) = [c_{(2)}(T)]^n$, and stroboscopic time evolution is of the form (15), and with $F(p) = e^{-iE_{1,p}T}c_{(2),p}(T)$.

Generally the condition (37) is **not** satisfied in the protocol we presented. We can see this writing the explicit expression for $d^*_{(2),p}$,

$$d^*_{(2),p} = -\frac{\left(E^2_{2,p} - E^2_{1,p}\right)}{4E_{1,p}E_{2,p}} e^{iE_{1,p}T_2} \sin(E_{2,p}T_2).$$ (39)

While it is possible to choose a duration $T_2$ such that the coefficient (39) vanishes for a given value of $p$, it is generally impossible for all of the coefficients with all values of $p$ to vanish simultaneously. This is why generally, the two-step driving protocol for the free massive boson does not generally lead to an integrable Floquet Hamiltonian. In the next subsection we find a similar result for a free fermionic system.

## 5.2 Free fermion

We consider the theory of a free, massive Majorana fermion. We will again perform a two-step diving protocol, where we alternate between the values of masses, $m_1$ and $m_2$, while demanding continuity of physical fields at each step. The corresponding Hamiltonians will again be denoted as $H_{m_1}$ and $H_{m_2}$.

The Majorana fermion corresponding to the system, $H_{m_1}$, can be described in terms of two component fields at $t = 0$,

$$
\begin{aligned}
\psi_+(x) &= \int dp\, e^{ipx} \left[ \alpha_{1,p}\, a_{1,p} + \alpha^*_{1,-p}\, a^\dagger_{1,-p} \right], \\
\psi_-(x) &= \int dp\, e^{ipx} \left[ \beta_{1,p}\, a_{1,p} + \beta^*_{1,-p}\, a^\dagger_{1,-p} \right],
\end{aligned}
$$ (40)

where $a_{1,p}$ and $a^\dagger_{1,p}$ are the anticommuting creation and annihilation operators, respectively, and we have defined the coefficients

$$\alpha_{1,p} = \frac{\omega}{2\pi\sqrt{2}} \frac{\sqrt{E_{1,p} + p}}{E_{1,p}}, \quad \beta_{1,p} = \frac{\omega^*}{2\pi\sqrt{2}} \frac{\sqrt{E_{1,p} - p}}{E_{1,p}},$$

with $\omega = \exp(i\pi/4)$, and dispersion relation $E^2_{1,p} = m^2_1 + p^2$. In terms of these creation and annihilation operators, the Hamiltonian can be written as

$$H_{m_1} = \int dp\, E_{1,p}\, a^\dagger_{1,p} a_{1,p}.$$ (41)

It will be useful to point out for future reference that this free Majorana fermion model can be obtained as the scaling limit of a transverse field quantum Ising chain (TFIC). The TFIC Hamiltonian is

$$H_{\text{Ising}} = -J \sum_{i=1}^{N} \left( \sigma^x_i \sigma^x_{i+1} + g\sigma^z_i \right),$$ (42)

where $\sigma^a$, with $a = x, y, z$, are Pauli matrices, and $J > 0$, and we impose periodic boundary conditions, $\sigma^a_{N+1} = \sigma^a_1$. The lattice spacing in (42) has been normalized to 1.

The Hamiltonian (42) can be diagonalized by performing series of simple transformations. This method is well documented in the literature (see [8] and references therein), so we only

mention the steps here without many details. First one transforms the spin operators into anticommuting fermionic ones by a Jordan-Wigner transformation,

$$c_i = \exp\left(i\pi \sum_{j<i} \sigma_j^+ \sigma_j^-\right) \sigma_i^-, \quad c_i^\dagger = \sigma_i^+ \exp\left(i\pi \sum_{j<i} \sigma_j^+ \sigma_j^-\right), \tag{43}$$

where $\sigma_i^\pm = \frac{1}{2}\left(\sigma_i^x \pm i\sigma_i^y\right)$. It is then useful to switch from position to momentum space by a Fourier transformation, $c_p = \frac{1}{\sqrt{N}} \sum_i^N c_i e^{ipi}$. The allowed values of momentum, $p$ depend on whether periodic or antiperiodic boundary conditions are imposed on the Fermions. We will focus here only on the antiperiodic sector, where $N$ is an even integer, and the momentum takes values $p_n = \frac{2\pi}{N}\left(n + \frac{1}{2}\right)$, an $n$ takes the values $n = -\frac{N}{2}, \ldots, \frac{N}{2} - 1$. The final step to diagonalize the Hamiltonian is performing a Bogoliubov transformation

$$a_p = u_p c_p - iv_p c_{-p}^\dagger, \quad a_p^\dagger = u_p c_p^\dagger + iv_p c_{-p},$$

with coefficients $u_p = \cos\frac{\theta_p}{2}$, $v_p = \sin\frac{\theta_p}{2}$, with $\theta_p$ fixed by

$$e^{i\theta_p} = \frac{g - e^{ip}}{\sqrt{1 + g^2 - 2g\cos p}}.$$

In terms of the operators $a_p$, $a_p^\dagger$, the Hamiltonian is

$$H_{\text{Ising}} = \sum_p E_p \left(a_p^\dagger a_p - \frac{1}{2}\right), \tag{44}$$

with dispersion relation

$$E_p = 2J\sqrt{1 + g^2 - 2g\cos p}. \tag{45}$$

The TFIC is known to have a transition at $g = 1$ between the ordered ($g < 1$) and disordered phases ($g > 1$). This is reflected in the spectrum given in (45), in that the particle-like excitations become gapless.

From the expressions (44) and (45), it is easy to see the free Majorana fermion can be obtained in the scaling limit, which is valid near the phase transition. This limit is obtained by re-introducing the lattice spacing, $\delta$, and then examining only the low energy degrees of freedom, while taking

$$J \to \infty, \quad g \to 1, \quad \delta \to 0,$$

while keeping the excitations' energy gap, $m$, and velocity, $c$, fixed. The dispersion relation (45) becomes, in physical units, $E_p = \sqrt{m^2 + (cp)^2}$, with

$$m = 2J|1 - g|, \quad c = 2J\delta,$$

which is the dispersion relation for a massive relativistic particle. In this scaling limit, the model defined by (44) is equivalent to the massive Majorana fermion defined by (41).

We now again consider the protocol where the initial state at $t = 0$ is some eigenstate of $H_{m_1}$, $|\Psi(0)\rangle$. At $t = 0$ we start driving with $H_{m_2}$, until time $T_2$, then switch back to the Hamiltonian $H_{m_1}$ until time $T$, and thus continue periodically driving.

Every time we switch between the two Hamiltonians, we demand continuity of both field components, $\psi_\pm(x, t)$, which allows us to write down Bogoliubov transformations that relate the new creation and annihilation operators with those of the previous step. We will not show

this calculation in detail, as it is similar to that presented in the previous subsection. At some time $t$, after which we have switched $n$ times between the two Hamiltonians, the creation and annihilation operators are related to those at $t = 0$ by the transformation

$$a^{(n)}_{[n]_2,p}(t) = c_{(n),p}(t) a_{1,p} + d_{(n),-p}(t) a^\dagger_{1,-p}, \qquad a^{\dagger(n)}_{[n]_2,-p}(t) = c^*_{(n),-p}(t) a^\dagger_{1,-p} + d^*_{(n),p}(t) a_{1,p},$$

where the Bogoliubov coefficients are time-evolved as

$$c_{(n),p}(t) = e^{-iE_{[n]_2,p}t} c_{n,p}, \qquad d_{(n),p}(t) e^{-iE_{[n_2],p}t} d_{(n),p}.$$

We can again derive a recursion relation between the coefficients at the $n$-th step, and those at the $(n-1)$-th step, by demanding continuity of the fields

$$
\begin{aligned}
c_{(n+1),p} &= e^{iE_{[n+1]_2,p}T_n} \left( \frac{\alpha_{[n]_2,p} c_{(n),p}(T_n) + \alpha^*_{[n]_2,-p} d^*_{(n),-p}(T_n)}{\alpha^*_{[n+1]_2,-p}} \right.\\
&\qquad \left. - \frac{\beta_{[n]_2,p} c_{(n),p}(T_n) + \beta^*_{[n]_2,-p} d^*_{(n),-p}(T_n)}{\beta^*_{[n+1]_2,-p}} \right)\\
&\qquad \times \left( \frac{\alpha_{[n+1]_2,p}}{\alpha^*_{[n+1]_2,-p}} - \frac{\beta_{[n+1]_2,p}}{\beta^*_{[n+1]_2,-p}} \right)^{-1},
\end{aligned}
$$

$$
\begin{aligned}
d^*_{(n+1),-p} &= e^{iE_{[n+1]_2,p}T_n} \left( \frac{\alpha_{[n]_2,p} c_{(n),p}(T_n) + \alpha^*_{[n]_2,-p} d_{(n),-p}(T_n)}{\alpha_{[n+1]_2,p}} \right.\\
&\qquad \left. - \frac{\beta_{[n]_2,p} c_{(n),p}(T_n) + \beta^*_{[n]_2,-p} d^*_{(n),-p}(T_n)}{\beta_{[n+1]_2,p}} \right)\\
&\qquad \times \left( \frac{\alpha^*_{[n+1]_2,-p}}{\alpha_{[n+1]_2,p}} - \frac{\beta^*_{[n+1]_2,-p}}{\beta_{[n+1]_2,p}} \right)^{-1},
\end{aligned}
\tag{46}
$$

with the notation for $[n]_2$ and $T_n$ we introduced in the previous subsection. With the recursive relations (46), we are able to compute correlation functions of $\psi_\pm(x,t)$ at any time.

As for the free Bosonic case, this simple two-step protocol does **not** lead to an integrable Floquet Hamiltonian. The requirement from Floquet integrability is again to have $d^*_{(2n),-p} = 0$, for all momenta, $p$, and integers, $n$. This condition is evidently not satisfied in general, as can be seen explicitly for $n = 1$,

$$
\begin{aligned}
d^*_{(2),-p} &= -\frac{i}{2E_{1,p}E_{2,p}} e^{iE_{1,p}T_2} \sin(E_{2,p}T_2)\\
&\qquad \times \left[ \sqrt{(E_{1,p}-p)(E_{2,p}+p)} - \sqrt{(E_{1,p}+p)(E_{2,p}-p)} \right]\\
&\qquad \times \left[ \sqrt{(E_{1,p}+p)(E_{2,p}+p)} + \sqrt{(E_{1,p}-p)(E_{2,p}-p)} \right].
\end{aligned}
\tag{47}
$$

We have seen in this section that in the two simplest cases of periodic driving we can consider, *i.e.*, two-step protocol for free bosonic and fermionic systems, integrable Floquet dynamics are not recovered. After $n$ full periods, the lack of elasticity is reflected in the fact that the Bogoliubov coefficients, $d^*_{(2n),p}$ do not vanish. In both, the bosonic and fermionic case, we see $d^*_{(2),p} \sim \sin(E_{2,p}T_2)$. While it is possible to find some particular time $T_2$ which will make a coefficient $d_{(2),p}$ vanish for a given $p$, it is generally not possible to ensure that these coefficients vanish for all values of $p$. In the next section we discuss several scenarios of driving protocols where integrability is preserved, and one can work around the problems raised in this section.

# 6 Examples of effective integrable Floquet field theory protocols

In this section we explore several approaches to preserve integrability under periodic driving. We have so far shown that if a driving protocol for a relativistic QFT leads to an integrable Floquet Hamiltonian, then the stroboscopic time evolution exhibits elasticity and factorization. Stroboscopic elasticity and factorization lead to several strong constraints on the time evolution of multi-particle states, as presented in Section 4. We then interpret the problem of finding an integrable driving protocol in QFT, to that of finding a driving protocol where all the constraints from Section 4 are satisfied.

We will start by examining two limits of periodic driving where integrability is trivially (approximately) preserved. The first of these is in the limit of very fast frequencies. As we will see, if our driving protocol consists of switching very frequently between two integrable QFT Hamiltonian, then the Floquet Hamiltonian is well approximated by the time-average of the two Hamiltonians. The second trivial limit where integrability is preserved consists of periodically driving by varying adiabatically slowly some parameter in an integrable QFT. We discuss how these limits lead to integrability in the free bosonic and fermionic theories.

We then discuss integrable Floquet protocols which rely on the concept of full revivals of states after a quantum quench. If one performs a quantum quench of a field theory in a finite volume, $L$, one expects that at some very long time, the state of the system returns to the initial state. If we precisely synchronize the driving frequency to the times when a full revival occurs, then this leads to stroboscopic elasticity and factorization. For a generic system, a full revival is expected to occur only at extremely large times which are exponential in terms of the system size. We explore, however, several situations where the time for a full revival can become much shorter. For field theories with massless particles, the time at which a full revival happens is proportional to the system size, and not exponential. We also present an even faster integrable driving protocol based on the quantum Ising chain with zero transverse field, where a full revival occurs even at infinite system size.

We finally discuss the different proposals for integrable Floquet dynamics that were presented in [19] by Gritsev and Polkovnikov. We show how the consequences of stroboscopic elasticity and factorization are compatible with some of their proposed driving protocols.

## 6.1 The trivially (approximately) integrable limits of the driving period

The first integrable driving protocols we consider correspond to two simple limits, corresponding respectively to very fast or very slow driving.

### 6.1.1 High-frequency protocols

When the driving period is very short, the Floquet Hamiltonian is simply given by the time-averaged Hamiltonian [36–40],

$$H_F = \frac{1}{T} \int_0^T dt H(t), \quad \text{(for } T \to 0\text{)}.$$

This result is simply recovered in the case of the two-step driving protocol described by (8), which is what we will consider for the rest of this subsection. The Floquet Hamiltonian can be written in terms of the BCH formula (9). In the limit $T \to 0$, the BCH reduces to

$$H_F = H_2 \frac{T_2}{T} + H_1 \frac{T_1}{T} + \mathcal{O}\left(\frac{T_1 T_2}{T}\right),$$

with $T = T_1 + T_2$, which is precisely the time-averaged Hamiltonian.

It is not difficult to choose two Hamiltonians $H_1$ and $H_2$ such that their average, therefore the Floquet Hamiltonian is integrable at small $T$. For example, we can choose them to be, as in the previous section, Hamiltonians corresponding to a free boson (or fermion) with different values of masses, $H_{m_1}$ and $H_{m_2}$.

As we have discussed in previous sections, integrability of the Floquet Hamiltonian implies stroboscopic factorization and elasticity. For free bosons and fermions, we found that this condition can be expressed in terms of the coefficients of the Bogoliubov transformation describing each step of the driving, as $d_{(2n),p} = 0$, for all momenta, $p$, and integer $n$.

We have computed these Bogoliubov coefficients corresponding to the periodic driving of a free boson (39) and free fermion (47). In both cases we computed explicitly the first coefficient and found

$$d_{(2),p} \sim \sin\left(E_{2,p} T_2\right).$$

It is then easy to see that for very short periods, with $T_2 \to 0$, we find $d_{(2),p} \to 0$. It is also similarly possible to show that $c_{(2),p} \to 1$ for $T_2 \to 0$. We can use these coefficients as initial conditions for the recursive relations (36) and (46), from which we will find $d_{(2n),p} \to 0$ as long as $T_2$ is small enough. It is then evident that in this case, as expected, integrability of $H_F$ at very short driving period implies stroboscopic elasticity and factorization.

If these high frequency Floquet Hamiltonians are integrable, then we should be able to describe the stroboscopic time evolution of a multi-particle state in terms of the function $F(\theta)$, which satisfies all the axioms presented in Section 4. As we previously discussed, in the case where $d_{(2),p} \approx 0$, then

$$F(\theta) \approx c_{(2),p},$$

where we discard contributions from $\mathcal{O}(T_2^2)$. Expanding Eq. (35) for the free boson, this is

$$F(\theta) \approx 1 + iT_2 \frac{E_{1,p}^2 - E_{2,p}^2}{2E_{1,p}} \approx \exp\left(iT_2 \frac{m_1^2 - m_2^2}{2E_{1,p}}\right), \tag{48}$$

where $E_{1,p} = m_1 \cosh\theta$, and $E_{2,p} = m_2 \cosh\xi$, with $\xi$ defined from the relation

$$p = m_1 \sinh\theta = m_2 \sinh\xi.$$

The function (48) satisfies all the axioms from Section 4.

For the free fermion driven with small $T$, we find from (46)

$$
\begin{aligned}
F(\theta) &\approx 1 + iE_{1,p}T_2 - i\left(\frac{\sqrt{(E_{1,p}^2 - p^2)(E_{2,p}^2 - p^2)} + p^2}{E_{1,p}}\right)T_2 \\
&\approx \exp\left[iE_{1,p}T_2 - i\left(\frac{m_1 m_2 + p^2}{E_{1,p}}\right)T_2\right].
\end{aligned}
\tag{49}
$$

The exponentiated expressions, (48) and (49) both satisfy all the axioms in Section 4.

### 6.1.2 Adiabatically slow driving

Integrability is also trivially approximately preserved in an opposite limit of driving speed, namely, for adiabatically slow driving protocols. The adiabatic limit is defined by very slowly varying some parameter in the Hamiltonian, such that at any point the system can be considered to be approximately at equilibrium.

In the case of the free boson and free fermion, we can consider a driving protocol of the type (10), where

$$H'(t) = H_{m(t)},\tag{50}$$

where the Hamiltonian on the right-hand side is that of a free boson, or fermion, with a time dependent mass, $m(t)$. The adiabatic limit is equivalent to the condition, $\frac{dm(t)}{dt} \approx 0$, for all times, $t$. We further ensure periodicity by requiring $m(0 + nT) = m(T_1 + nT)$, for integers $n$.

If the initial state is chosen to be the ground state of $H_{\text{int}}$, then we assume that if the driving is adiabatically slow, then one expects that at any future time, $t$, the system is in the ground state of the current Hamiltonian, $H(t)$. The statement can be generalized to excited states containing particles, where adiabatically slow driving does not change the particle content (but it changes the state's energy, since the particle mass changes). This then implies that stroboscopic integrability is preserved under adiabatic periodic driving, since after a full period, the particle content of the state is conserved

Under these assumptions, we can determine that there is a function $F(\theta)$ describing the integrable stroboscopic time evolution, which is given by

$$F(\theta) = \exp\left( iT_1 E_p - i \int_0^{T_1} dt E_p(t) \right),$$

where $E_p = m(0)\cosh\theta$, and $E_p(t) = m(t)\cosh\xi(t)$, where $p = m(0)\sinh\theta = m(t)\sinh\xi(t)$.

## 6.2 Integrability based on full revivals

As we have seen explicitly, the two-step driving protocols for free bosons and fermions do not yield integrable Floquet dynamics. We have shown that the condition of stroboscopic elasticity is explicitly broken by the fact that the coefficients $d_{(2),p} \sim \sin(E_{2,p} T_2)$ do not generally vanish for all values of $p$. It is always possible to chose some time $T_2 = (\pi/2 + n\pi)/E_{2,p}$ with integer, $n$, such that $d_{(2),p} = 0$ for a specific $p$, however, the coefficient will not generally vanish for other values of $p$.

The integrable protocols we will discuss in this section consist on allowing enough time between driving steps so that the system experiences a full revival. If at time $t = 0$ we perform a quantum quench by changing some parameter in the Hamiltonian (for example, the mass of the free boson/fermion), there is a full revival if at some time $t = T_R$ the state of the system completely returns to the initial state, $|\Psi(T_R)\rangle = |\Psi(0)\rangle$. If we precisely tune the two-state driving protocol, such that at time $T_R$, we switch back to the original Hamiltonian, then integrability will be preserved. In terms of our expressions for $d_{(2),p}$, the revival time $T_R$, if it exists, is defined such that $\sin(E_{2,p} T_R) = 0$ for all $p$. In the following subsections we will discuss the conditions under which such complete revivals are possible.

Under these revival-based integrable driving protocols, the stroboscopic time-evolution of particles is described by the function

$$F(\theta) = e^{iT_R E_{1,p}}.\tag{51}$$

### 6.2.1 Exponentially-slow revival protocol for generic finite systems

A measure of how close the state at time $t$ is to the initial state is given by the return amplitude,

$$l(t) = |\langle\Psi(0)|\Psi(t)\rangle|,$$

where the initial state is normalized such that $l(0) = 1$. When a full revival occurs, at time $T_R$, then the return amplitude is again $l(T_R) = 1$. It is also possible that a full revival never occurs, but there are approximate revivals at times when $l(t) \approx 1$, when the state is arbitrarily close to the initial state.

A full or approximate revival is expected to occur generically when there is a finite system size, $L$, and a high-momentum cut-off, provided, for instance by placing the system on a lattice. In a finite system, the allowed particle momenta become quantized. For the free bosonic and fermionic systems we consider, the quantization condition is given by $p = 2\pi n/L$, where $n$ takes integer values for bosons and half-integer for fermions. For integrable interacting field theories, the quantization conditions are more complicated, depending on the number of particles in the given state and the S-matrices between the particles [18]. If the system is regularized by an underlying lattice, then there is also a high-momentum cutoff, such that there is a finite number of possible momentum modes.

Once we have a regularized system at finite volume, an approximate (but arbitrarily close to full) revival is guaranteed by the quantum version of the Poincare recurrence theorem [41]. The statement of this theorem is that when the spectrum of a given system is discrete, and bounded, there always exists some revival time, $T_R$, such that

$$1 - l(T_R) < \delta,$$

for any arbitrarily small number $\delta$. If we perform a two step driving protocol as described for a free boson or fermion in Section 5, such that at the revival time, $T_R$, we switch back to the original Hamiltonian, then this implies that $d_{(2),p} \approx 0$ for all $p$. This implies that it is always possible to, at least approximately, preserve integrability by precisely tuning the driving period with the recurrence time.

The downside of such a protocol is that revival times, $T_R$ are expected to be extremely long. In a generic system, the recurrence time is expected to grow exponentially with the system size, such that $T_R \sim e^{\alpha L}$. In the next subsections we will study specific scenarios where the revival time can be significantly shorter.

### 6.2.2 Linearly-slow revival protocols in massless field theories

While we have discussed that it is generally possible to preserve Floquet integrability by precisely tuning a two-step driving protocol to a given revival time, this is a very impractical protocol to carry out, since the revival time is expected to grow exponentially with the system size. We are now interested in exploring specific driving protocols that may result in a much shorter and practical revival time.

The simplest way to drastically reduce the revival time is to consider the time evolution of a massless relativistic QFT in a finite system size, $L$. The reason revivals occur much faster for a massless model is that due to relativistic invariance, all particles must travel at the same speed *i.e.* the speed of light, $c$ (throughout this paper we have used the standard convention, $c = 1$).

In this case, the revival time is expected to become linearly proportional to the system size, instead of growing exponentially, meaning that this is a much more realistic driving protocol to perform. The initial state after a quantum quench can be expressed as a superposition of multi-particle states of the post-quench Hamiltonian. The state of the system at some time, $t$, is obtained by allowing the particles to propagate.

The idea behind the short-time revival in massless models, is that since all the particles propagate at the same speed, in a closed system with some sort of periodic boundary conditions, all particles return to their initial positions at the same time. For a free bosonic system of size $L$, with periodic boundary conditions, the system returns to its initial state after all the particles have circled around the whole system once. The revival time is then $T_R = L/c$.

In a free fermionic system, antiperiodic boundary conditions are usually imposed. This means that particles need to go around the full circle twice for the system to return to its initial state, yielding a revival time $T_R = 2L/c$. It has been found that revival times linear in $L$ are a generic feature of quantum quenches in CFT [42]. As with the bosonic and fermionic cases, the massless particles need to go around the full circle some integer number of times, depending on the necessary boundary conditions.

These results can also be understood in terms of the Bogoliubov coefficients, $d_{(2),p}$, which we expect to vanish for an integrable quench. As we have discussed, for both, free bosons and fermions, this coefficient is proportional to $\sin(E_{2,p}T_2)$. In a finite volume, momentum is quantized as $p = 2\pi n/L$, with $n$ taking integer or half integer values, for bosons or fermions, respectively. If the particles are massless, then the energies are also quantized as

$$E_{2,p_n} = c\frac{2\pi n}{L},$$

It is then easy to see that if we tune the driving period to be the revival time $T_2 = L/c$ for bosons, or $T_2 = 2L/c$ for fermions, then we have

$$d_{(2),p} \quad \sim \quad \sin(2\pi n) = 0, \quad \text{(bosonic)}$$

$$d_{(2),p} \quad \sim \quad \sin(4\pi n) = 0, \quad \text{(fermionic)},$$

for all $p$, as is necessary to preserve Floquet integrability.

### 6.2.3 Systems with homogeneous spectrum

It is possible to reduce even further the time to reach a full revival if we consider specific models with even more peculiar spectrum properties. The fastest such case is when we consider a theory whose excitations have a homogeneous spectrum, which does not depend on the particle's momentum. In such a system, the energy carried by an excitation is some constant $E_p = E$, regardless of momentum.

While this is a rare and restrictive scenario, it is how the spectrum of the transverse field Ising chain behaves in two different extreme limits. We can split the TFIC Hamiltonian (42) in two terms, $H_{\text{Ising}} = H_{\text{Ising}}^1 + H_{\text{Ising}}^2$, such that

$$H_{\text{Ising}}^1 = -J\sum_{i=1}^{N}\sigma_i^x\sigma_{i+1}^x, \qquad H_{\text{Ising}}^2 = -Jg\sum_{i=1}^{N}\sigma_i^z.$$

In the special limits where one of the two terms of the Hamiltonian strongly dominates over the other, the particle spectrum becomes homogeneous. This can be seen from the explicit expression for the particle dispersion relation

$$E_p = 2J\sqrt{1 + g^2 - 2g\cos p}.$$

The limit where $H_{\text{Ising}}^1$ dominates is given by $g \to 0$, for which the particle spectrum is given by $E_p = 2J$.

The limit where $H_{\text{Ising}}^2$ dominates is given by taking $J \to 0$ and $g \to \infty$, while keeping $Jg$ constant. In this case, the particle spectrum is given by $E_p = 2Jg$.

In both cases, the total energy of a given state depends only on what is the total number of excited particles, and not on what are the individual momenta of the particles.

We now consider the two step protocol for the free fermionic system, where we alternate between the full Hamiltonian $H_{\text{Ising}}$ (for which we can consider the scaling limit), and the

second Hamiltonian is either $H_{\text{Ising}}^1$ or $H_{\text{Ising}}^2$. We consider the initial state to be some eigenstate of $H_{\text{Ising}}$. We then at time $t = 0$ switch to the second Hamiltonian, let the system evolve, and at $t = T_2$ switch back to the original full TFIC Hamiltonian. We again can compute Bogoliubov coefficients relating the creation and annihilation operators at different times, such that $d_{(2),p} \sim \sin(E_{2,p} T_2)$.

The main simplification in this case is that, since $E_{2,p}$ is a constant given by either $E_2 = 2J$, or $E_2 = 2Jg$, then we can simply choose the driving period $T_2 = n\pi/E_2$ for some integer $n$, which will ensure $d_{(2),p} = 0$, preserving integrability.

In this case, all the particles in some given state are time-evolved with the same phase, $e^{-iE_2 t}$, so a full revival, regardless of the initial state, will be reached every time this factor equals one, or $t = n2\pi/E_2$.

We note that in this system, a full revival occurs even at infinite system size, which means this integrable driving protocol can be performed much faster than those discussed in the previous subsections.

It is important to point out that in this section we have identified a simple integrable driving protocol by searching for the conditions of stroboscopic elasticity and factorization. We have not explicitly examined what is the corresponding Floquet Hamiltonian, or its associated conserved charges. The exact form of the Floquet Hamiltonian is not necessarily easy to find in closed form from the BCH expansion, (9), since the period $T_2 = n\pi/E_2$ could be large, not supporting such a perturbative expansion. At this point, in fact, we do not have any simple proposal for computing the Floquet Hamiltonian corresponding to this protocol, but one of the great advantages of the formalism developed in this paper is that we do not need to find this Hamiltonian! It is sufficient for us to identify that the properties of elasticity and factorization are present, and find the corresponding $F(\theta)$ function, given in this case in Eq. (51). This approach parallels that of ordinary integrable QFT's, when one can often work without knowledge of the system's Hamiltonian, and the problem is instead shifted to finding solutions to the S-matrix axioms, reflecting elasticity and factorization.

## 6.3 The Gritsev-Polkovnikov scenarios

The study of integrable Floquet protocols was advanced by Gritsev and Polkovnikov in Ref. [19]. In this reference, the authors proposed several different definitions for what could be interpreted as Floquet integrability. In certain cases full agreement can be shown with our definition, where an infinite set of local charges which commute with the Floquet Hamiltonian were found. In practice, integrable Floquet systems were defined in [19] as systems which "do not heat up" with time under periodic driving. This is a much weaker restriction than our definition of Floquet integrability, since only the overall temperature needs to be a conserved quantity. In the next subsections we will find under what circumstances are the Gritsev-Polkovnikov integrable driving protocols compatible with our results.

### 6.3.1 Onsager algebra two-step process and row-transfer matrix protocols

If one is interested in periodically-driven systems which do not "heat up", it is sufficient (but not necessary) to require that the Floquet Hamiltonian remains local. The eigenstates of a highly non-local Hamiltonian may be indistinguishable from high-temperature states of a local Hamiltonian, which is why driven systems are generally expected to heat up [20]. If one is able to find some driving protocol such that the Floquet Hamiltonian is local, then traditional energy conservation will prevent heating.

One proposal presented in [19] was to study two-step driving protocols, where the two Hamiltonians, $H_1$ and $H_2$, satisfy specific algebraic properties that lead to locality of $H_F$. Particularly one can consider the two Hamiltonians to be linear combinations of generators of

some Lie group, $Q_a$, which satisfy the algebra

$$[Q_a, Q_b] = f_{ab}^c Q_c, \tag{52}$$

where $f_{ab}^c$ are structure constants, and $a, b, c$ are labels whose range depends on the dimension of the group. The Floquet Hamiltonian can then be obtained from the BCH formula (9), which involves a sum of nested commutators of $H_1$ and $H_2$. Given the relation (52), each of the nested commutators can itself be expressed as a linear combination of the same generators. The ultimate consequence is that the Floquet Hamiltonian itself is a linear combination of these generators, keeping the same locality properties as the original Hamiltonians, $H_1$ and $H_2$.

The concept of Hamiltonians which are Lie-group generators does not seem immediately to be related to quantum field theory, or any other extended (1+1)-d systems. In [19], however, the authors argued that the same ideas can also be applied to some infinite-dimensional Lie algebras. The relevant example to us is the driving protocol based on the Onsager algebra.

The Onsager algebra is defined by the two "seed" operators

$$A_0 = \sum_{i=1}^{L} \sigma_i^z, \qquad A_1 = \sum_{i=1}^{L} \sigma_i^x \sigma_{i+1}^x, \tag{53}$$

which can be recognized as the terms that define the TFIC Hamiltonian, (42). These operators generate the infinite dimensional algebra of operators $A_n, G_n$, with integer index $n$, satisfying

$$
\begin{aligned}
{[A_l, A_m]} &= 4G_{l-m}, \quad l \geq m, \\
{[G_l, A_m]} &= 2A_{m+l} - 2A_{m-l}, \\
{[G_l, G_m]} &= 0.
\end{aligned}
$$

In this case it was proposed that the two-step driving protocol based on the two Hamiltonians $H_1 = -\alpha A_0$ and $H_2 = -\beta A_1$, for constants $\alpha, \beta$, leads to an integrable Floquet Hamiltonian. It was indeed found that there exist an infinite number of local charges that commute with $H_F$. Naively then we should expect to find stroboscopic elasticity and factorization as we have proposed, however, this is not the case since there is one crucial property not satisfied by the conserved charges. There is not a subset of conserved charges which are *parity-even*.

To see this we briefly recall what are the conserved charges of the Floquet Hamiltonian, which were originally found in [43, 44]. We first introduce the notation from Ref. [43, 44]

$$Z_{[s_1 s_2 \ldots s_p]} = \sum_{i=1}^{N} \sigma_i^{s_1} \sigma_{i+1}^{s_2} \ldots \sigma_{i+p-1}^{s_p},$$

where the label $s \in \{1 = x, 2 = y, 3 = z\}$ identifies one of the three Pauli matrices. With this notation, we define the operators

$$
U_n = \begin{cases}
Z_{[1(3^{n-1})1]}, & n \geq 1, \\
-Z_{[3]}, & n = 0, \\
Z_{[2(3^{-n-1})2]}, & n \leq -1,
\end{cases}
$$

$$
V_n = \begin{cases}
Z_{[1(3^{n-1})2]}, & n \geq 1, \\
1, & n = 0, \\
-Z_{[2(3^{-n-1})1]}, & n \leq -1,
\end{cases}
$$

where $n$ is an integer.

The infinite set of local conserved charges was explicitly found for this driving protocol, and are given by the two sets [43, 44]:

$$
\begin{aligned}
C_n &= V_{n+1} + V_{-n-1}, \\
Q_n &= A(U_{n+1} + U_{-n+1}) + B(U_n + U_{-n}) - C(V_{n+1} + V_{-n+1} - V_{n-1} - V_{-n-1}),
\end{aligned}
$$

for integers, $n$, and constants

$$
\begin{aligned}
A &= \cos(2\beta T_2)\sin(2\alpha T_1) \\
B &= \sin(2\beta T_2)\cos(2\alpha T_1) \\
C &= \sin(2\beta T_2)\sin(2\alpha T_1).
\end{aligned}
$$

We point out that the charges $C_n$ and $Q_n$ are not even under spatial reflections, such that the there is not an infinite set of conserved charges with positive-definite eigenvalues. This lack of parity-even charges comes from the fact that they all involve the operators $V_n$, unlike the usual conserved charges of the Ising chain, where parity-even charges are written in terms of $U_n$ operators only. It is then clear that we cannot isolate conserved charges containing only $U_n$ operators, therefore we do not necessarily expect elasticity or factorization.

We do note, however, for the specific scenario where we choose the period such that either $2\beta T_2 = k\pi$, or $2\alpha T_1 = k\pi$, for some integer, $k$, the constant, $C$, vanishes. This renders the charges $Q_n$ parity-even, so we should expect elasticity and factorization. This scenario, is actually just the full revival protocol we have discussed in the previous section, where the Hamiltonian is switched at precisely the moment the system returns to its initial state. We have already shown stroboscopic elasticity and factorization appear in this scenario.

We can test explicitly the properties of the driving protocol we have described, by explicitly computing the Bogoliubov coefficients which connect the creation and annihilation operators at different times, as we have done for the free bosonic and fermionic field theories. The new Bogoliubov coefficients are derived in analogy with what we have done in Section 5, by demanding that the Jordan-Wigner fermions, $c_i$ from Eq. (43) be continuous at each driving step. Assuming the initial state is some eigenstate of $H_1$, we can see if stroboscopic elasticity and factorization are present by computing the Bogoliubov coefficient $d_{(2),p}^*$, as was done for the free bosonic and fermionic fields. This can be computed explicitly to find

$$
d_{(2),p}^* = \mathrm{i} e^{\mathrm{i}2\alpha T_2}\sin(2\beta T_2)\sin\left(\frac{p+\pi}{2}\right)\cos\left(\frac{p+\pi}{2}\right).
$$

After a full period, $T$, the system will then not be in the same eigenstate of $H_1$ as initially, so stroboscopic elasticity is broken. We note, however, that if the period is chosen such that $2\beta T_2 = k\pi$, then Floquet integrability is recovered.

Another similar relevant driving protocol was proposed in Ref. [19], based on considering the two-step process Hamiltonians, $H_{1,2}$ to be given by the row and column transfer matrix of a classical integrable spin model, respectively. We will not explore deeply this scenario, but only point out the two main concrete examples that were proposed in [19]. When the transfer matrix of the Ising model is considered, the Hamiltonians arising from the row and column transfer matrices are just the same operators $A_0$ and $A_1$ we have already discussed.

A second proposed example in [19] is given by considering the transfer matrix corresponding to the $XXX$ Heisenberg spin chain, whose Hamiltonian is given by

$$
H^{(XXX)} = \sum_i U_i,
$$

with

$$
U_i = -\frac{1}{2}\left[\sigma_i^x \sigma_{i+1}^x + \sigma_i^y \sigma_{i+1}^y + \sigma_i^z \sigma_{i+1}^z - 1\right].
$$

From the corresponding row and column transfer matrices, it is proposed in [19] that the two-step process given by the alternating between the two Hamiltonians

$$
H_1^{(XXX)} = \sum_i U_{2i}, \quad H_2^{(XXX)} = \sum_i U_{2i+1}, \tag{54}
$$

leads to an integrable Floquet Hamiltonian. It is beyond the scope of this paper to verify that the constraints from stroboscopic elasticity and factorization apply to the Floquet Hamiltonian generated by the two-step protocol based on (54). This will be much more difficult to verify explicitly than for the Ising chain case, because the *XXX* spin chain is an interacting model which means the creation and annihilation operators at different times are not related by Bogoliubov transformations, making the simple techniques we have developed for free models inapplicable.

Even though at this moment we cannot explicitly examine the *XXX* Floquet protocol just described to the level of detail of the Ising chain protocol, we point out that it seems likely that it can be shown in the future to satisfy our criteria for integrability. Unlike the Ising protocol we have studied in this section, it is possible that in the *XXX* case, the Floquet Hamiltonian does have a set of parity-even conserved charges, as it is seen in [19] at least at the level of the first few terms of the BCH formula. The first few terms of $H_F$ in this expansion are proportional to the standard conserved charges of the *XXX* chain, such that it can be easily shown to have parity-even conserved charges. If the higher orders of the BCH expansion are shown to also commute with these parity-even charges then our criteria for integrability will be satisfied.

### 6.3.2 Boost-operator based protocols

We now examine the last type of integrable driving protocol that was proposed in [19], which is based on the so-called "boost"operator. Such an operator was defined in [19] in terms of the transfer matrices of integrable lattice models as one that generates a shift in the spectral parameter. Since we are mainly interested in relativistic field theories in this paper, we will study the characteristics of the Boost operator in the scaling limit. In the relativistic scaling limit, the Boost operator, denoted as $B$ simply corresponds to the generator of a Lorentz boost, satisfying the Poincare algebra, together with the Hamiltonian and total momentum operator,

$$[H,P] = 0, \quad [B,P] = iH, \quad [B,H] = iP. \tag{55}$$

We can consider a two-step driving protocol, where one of the Hamiltonians is that of an integrable QFT, $H_1 = H_{\text{int}}$, and the second Hamiltonian is the corresponding boost operator $H_2 = B$. The initial state is assumed to be some eigenstate of $H_{\text{int}}$, such that $|\Psi(0)\rangle = |\{\theta_i\}\rangle$.

It was proposed in Ref. [19] that this driving protocol is integrable, according to their definition. We point out, however, that this protocol only satisfies a very weak definition of integrability, in that the system "does not heat up" upon driving. The action of the boost operator on the initial state is given by

$$e^{i\alpha B}|\{\theta_i\}\rangle = |\{\theta_i + \alpha\}\rangle,$$

such that all the particle rapidities are shifted by the same amount, $\alpha$. While this action does increase the total energy of the state, since all the particles are now moving faster, the boost does not increase the entropy of the system. The energy input is organized, and the system remains in a pure eigenstate of $H_{\text{int}}$ upon driving, therefore the temperature of the system does not increase, yet the energy does.

Such a two-step protocol is not integrable as defined in this paper. There is no stroboscopic elasticity and factorization, since the momenta of particles are not conserved in the time evolution.

Another driving protocol based on the boost operator was proposed in [19] which does in fact satisfy our definition of Floquet integrability under certain conditions. These were called "quantum boost clocks". We consider a driving protocol given by the Hamiltonian

$$H(t) = H_{\text{int}} + b(t)B, \tag{56}$$

where $H_{\text{int}}$ is an integrable QFT, and where $b(t)$ is some periodic function $b(t + T) = b(t)$. It was shown in [19] that these protocols are integrable when the time average of $b(t)$ is zero [2],

$$\bar{b} \equiv \frac{1}{T} \int_0^T b(t) dt = 0.$$

To show integrability, we switch to a rotating reference frame, generated by the unitary operator

$$
\begin{aligned}
V(t) &= \exp(-i\mathcal{F}(t)B), \\
\mathcal{F}(t) &= \int_0^t \delta b(t') dt'.
\end{aligned}
\tag{57}
$$

where $\delta b(t) \equiv b(t) - \bar{b}$. The Hamiltonian in the rotating frame is

$$
\begin{aligned}
H_{\text{rot}}(t) &= V^\dagger(t) H V(t) - iV^\dagger(t) d_t V^\dagger = \bar{b} B + V^\dagger(t) H_{\text{int}} V(t) \\
&= \bar{b} B + \sum_{n=1}^\infty \frac{[\mathcal{F}(t)]^{n-1}}{(n-1)!} Q_n,
\end{aligned}
\tag{58}
$$

where for an integrable QFT $Q_n = H_{\text{int}}$ for odd $n$, and $Q_n = P$ (the total corresponding momentum operator), for even $n$. The expressions for the charges $Q_n$ are more involved in general integrable lattice models [19], but reduce to only the Hamiltonian and momentum operators in the continuum.

Integrability emerges when we require $\bar{b} = 0$. The first term in the right hand side of the rotating frame Hamiltonian (58), which is proportional to the boost operator, vanishes. This means that all the individual terms in the Hamiltonian commute with each other. In this case the Floquet Hamiltonian can be written as the time-averaged rotating frame Hamiltonian,

$$H_F = \frac{1}{T} \int_0^T dt\, H_{\text{rot}}(t).
\tag{59}$$

In the case that $\bar{b} = 0$ the terms in $H_F$ proportional to the momentum operator vanish as well. This is because the time-average of an odd power of the function $\mathcal{F}(t)$ is zero. As we will see, the vanishing of these momentum-operator terms is necessary for integrability, otherwise some of the axioms presented in Section 4 would not be satisfied.

In the $\bar{b} = 0$ case, the Floquet Hamiltonian is then

$$H_F = \sum_{n=0}^\infty \frac{\bar{\mathcal{F}^{2n}}}{(2n)!} H_{\text{int}},
\tag{60}$$

where

$$\bar{\mathcal{F}^{2n}} = \frac{1}{T} \int_0^T dt\, [\mathcal{F}(t)]^{2n}.$$

---

[2] In general lattice models there are other values of $\bar{b}$ which lead to integrable Floquet dynamics. This is related to the fact that the particle energy spectrum for a discrete system, such as the TFIC, can be a periodic function of the particle momentum. When one performs a boost with the operator $\exp(i\alpha B)$, the momentum is increased, but the energy will be a periodic function of $\alpha$. On the other hand, for QFT's in the continuum limit, a boost always increases the energy, as well as the momentum, so no periodic behavior exists, which leads to $\bar{b} = 0$ being the only integrable point, in contrast with other examples found in [19].

If $H_{\text{int}}$ is an integrable QFT Hamiltonian, whose spectrum consists of one particle with mass $m$, then we can easily read off from (60) that there is stroboscopic elasticity and factorization, described by the function

$$F(\theta) = \exp\left\{-\mathrm{i}T\left[\left(\sum_{n=0}^{\infty}\frac{\bar{\mathcal{F}}^{2n}}{(2n)!}\right) - 1\right]m\cosh\theta\right\}, \tag{61}$$

which does satisfy all the axioms from Section 4. We point out that, if there had been any terms proportional to $P$ in the Floquet Hamiltonian, then the function $\log(F(\theta))$ would include terms proportional to $\sinh\theta$, in addition to the $\cosh\theta$ terms. This would lead to the violation of the Floquet cross-unitarity axiom (19).

## 7 Conclusions

We have presented a concrete definition of integrability that applies to periodically driven quantum field theoretical models. In particular, we elucidated what is the computational advantage of having integrability, and how it can be associated with powerful analytical tools.

Integrable Floquet Hamiltonians are those which possess the properties of stroboscopic elasticity and factorization. We showed that these properties arise when there is an infinite number of independent local charges which commute with the Floquet Hamiltonian, and an infinite subset of them are positive definite.

Once it has been established that stroboscopic elasticity and factorization are present in some driving protocol, it is possible to write down a set of axioms that greatly constrain the stroboscopic time evolution of particles. This is done assuming that for some portion of the period the model is evolved with the Hamiltonian of a known integrable QFT, then the effect of driving can be expressed as a modification in the time evolution of the particles of this Hamiltonian, by dressing them with some function, $F(\theta)$. Our new set of axioms are analogous to the usual axioms in standard integrable field theory, which constrain the form of the S-matrix [4], and the axioms in integrable boundary field theory, which constrain the form of the particle reflection matrix [27].

The majority of driving protocols are not integrable. Integrable Floquet Hamiltonians may arise only under certain controlled conditions. We showed that some of the simplest driving protocols one can design, namely free bosonic and free fermionic field theories, where the value of the particle mass is periodically alternated, do not generally lead to an integrable Floquet Hamiltonian. Given the simplicity of these models, the effects of periodic driving can be computed analytically, and one can see explicitly that the particle content of the system may change stroboscopically, violating stroboscopic elasticity.

We then proceeded to discuss a set of simple protocols where integrable Floquet dynamics can be observed. The first of these corresponds to an approximation of the Floquet Hamiltonian for very short driving period, where the Floquet Hamiltonian can be written as the time-averaged Hamiltonian over the period. In this case it is easy to choose time-dependent Hamiltonian such that the time-average is integrable. This property of short-period driving is well known previously to this paper [36–40]. We show however, how this approximate integrability is compatible with our formalism and the set of axioms we introduce. Particularly we compute the dressing function $F(\theta)$ corresponding to very fast driving in free bosonic and fermionic systems.

Another simple integrable case we discussed is when the system is driven adiabatically slow. In the case corresponding to a free bosonic or fermionic system where the mass is varied adiabatically slowly, it is also simple to write down the corresponding $F(\theta)$ function.

Floquet integrability was also seen to arise in protocols that are finely tuned to a full revival in the system. After a quantum quench, a system is in a very far from equilibrium state, which will be subject to time evolution. Under certain conditions it is possible that the system will return to the same initial state after some amount of time; if at this point one performs another quench back to the original Hamiltonian, then the stroboscopic dynamics will be integrable. For a generic system an approximate revival is expected when the system is placed in a finite volume, however, the time when a this approximate revival is expected grows exponentially with system size, so it is not a very realistic protocol. The situation can be improved in field theories with massless particle excitations. In this case a full revival is expected to occur at a time that is linearly proportional to the system size, which is much more practical for attempting to realize this protocol. The situation can be even further improved for models with a special spectrum, where any particle has the same energy, regardless of its momentum, such as is the case with the quantum Ising chain with no transverse field. In this case a full revival happens in a time-scale inversely proportional to the particle energy, and is possible even at infinite system size.

We then considered some of the protocols that were proposed as integrable by Gritsev and Polkovnikov in [19]. One interesting proposed protocol consisted of alternating periodically between the two terms in a transverse field Ising chain Hamiltonian (53). As was first pointed out in [43,44], this protocol seems to yield an integrable Floquet Hamiltonian, in the sense that one can write down an infinite set of local charges that commute with it. We showed, however, that this protocol does not satisfy the strict definition of integrability presented here, and there is no stroboscopic elasticity. This can be understood from the fact that even though there is an infinite number of conserved charges, these are not positive definite, so our derivation of stroboscopic elasticity and factorization does not need to apply.

Finally we discussed the integrable driving protocols based on the boost operator, proposed in [19]. In this case we show that the "quantum boost clock" protocol, described here in Eq. (56), leads to an integrable Floquet Hamiltonian, fully compatible with our proposed properties of stroboscopic elasticity and factorization. We then computed the corresponding function $F(\theta)$ which describes the time evolution of particles (61).

While we have shown that some protocols leading to an integrable, or approximately integrable Floquet Hamiltonian do exist, they remain generally very special and finely tuned protocols. The protocols exhibiting integrable Floquet dynamics studied in this paper mostly concern free bosonic and fermionic systems, with very simple results. We point out, however that this is merely a reflection of our inability at this point to perform similar analytic calculations for nontrivial interacting models. There does not seem to be any fundamental reason prohibiting less trivial interacting integrable Floquet systems, and we expect more such examples will be discovered in the future. The list of integrable protocols presented here is not necessarily exhaustive. It would be very interesting in the future to understand if there is any possible classification or systematic way of obtaining new driving protocols leading to integrable Floquet Hamiltonians.

As a speculative outlook, we can propose a few examples of protocols which we believe are good candidates to search for integrability in the future. We will mention three different protocols where we believe Floquet integrability may be found.

The first protocol consists on two-step protocols involving quantum quenches which do not change the ground-state of the system. One particular example consists of considering a sinh-Gordon model, with potential,

$$V(\phi) = \frac{m_0^2}{g^2} \cosh g\phi,$$

and alternating between two different values of the coupling constant, $g$, while keeping the

physical particle mass constant. This kind of constant-mass quenches leave the energy spectrum of particles unchanged (in the infinite volume limit). The initial states corresponding to some quenches in sinh Gordon have been computed in [**?**, 45]. It can be seen from the result of [**?**, 45] that one does not create particles after the quench if only the coupling constant is changed, but not the physical mass of particles. It then seems possible that if one performs a periodic driving protocol alternating between values of $g$, that it may exhibit stroboscopic elasticity. Such conservation of particle number is seen explicitly in [47] for some such quantum quenches which do not change the ground state of the field theory.

A second protocol which might be shown to display integrability is what we could call a "quantum dilatation clock", in analogy to the quantum boost clock protocols defined in [19]. This protocol would consist on starting with a CFT at finite volume, prepared initially in one of its eigenstates, then performing a periodic protocol analogous to Eq.(56), but replacing the boost operator, $B$, with the CFT's corresponding dilatation operator, which we can call $D$. This protocol would correspond to periodically stretching and shrinking the given CFT. Geometrically, this protocol would correspond to computing the partition function of the CFT on a geometry of an undulating "cylinder", where the radius of the cylinder periodically varies along the longitudinal direction. Since this geometry can be obtained as a conformal map from a standard cylindrical geometry, we expect that integrability properties of the CFT will not be broken. We point out, that after the publication of this preprint, a study of a very similar protocol to the one proposed here was published in [48], where a driving protocol was obtained by performing some particular conformal transformation of a CFT. It was indeed shown that there was a "non-heating phase" where no energy is absorbed with driving. In fact the entanglement entropy is computed and shown to be time-periodic in this phase. These two results seem to be consistent with our definition of integrability.

Finally one particularly promising candidate for Floquet integrability concerns the recently studied cases of Floquet quantum criticality [49–51]. It was shown there exist some critical points describing phase transitions even in periodically driven systems. It is well known that in equilibrium (1+1)-dimensional field theory, a critical point, which displays conformal invariance implies an infinite-dimensional symmetry [52]. This results in (1+1)-dimensional CFT's being also integrable. It would be very interesting to see if such analogous infinite symmetry can be shown for Floquet critical systems, which would imply that these systems are also Floquet integrable. To proceed in this direction, one would need to better understand the properties of criticality in driven systems, particularly to study the divergence of correlation lengths at these points. It is well known that scale invariance, plus Poincaré invariance imply conformal symmetry in QFT [53], which in 2d is an infinite dimensional symmetry. One would then need to explore what are consequences of scale invariance, and a stroboscopic, discrete reduction of Poincaré invariance. Ideally it would be possible to show that scale invariance in the driven system still implies some larger conformal-like symmetry which could be shown to be infinite dimensional in 2d.

We point out that stroboscopic elasticity implies a periodic structure in time, which is reminiscent of the recently studied "quantum time crystals" [54,55]. Floquet integrability prevents the sets of particle rapidities from changing stroboscopically, which means the system exhibits a periodic structure in time, while breaking continuous time translation invariance. It would be interesting to understand in the future if there are any deeper connections that can be made between Floquet integrability and the time crystals literature.

In standard integrable QFT, it is possible to define new integrable field theories by looking for S-matrices which are new solutions of the corresponding axioms with some specified symmetry properties, without referring to the system's Hamiltonian. It would be ideal in the future if some similar approach can be established to study integrable driven systems. One would search for $F(\theta)$ functions which are new solutions to the axioms presented here, and

define the driven system in terms of this solution. In the end a significant goal would be to identify what is the corresponding Floquet Hamiltonian which yields this new $F(\theta)$ function.

An interesting possible application of our results in the future would be to develop some kind of perturbative formalism to analytically study small deformations of integrable Floquet Hamiltonians. While our results of this paper are limited to integrable Floquet Hamiltonians, which seems to be a very strong restriction, it may be possible that a wider set of physical driving protocols can be described as not integrable, but perturbatively close to an integrable protocol. This program would be similar to the approach developed to study deformations of integrable QFT's using knowledge of the exact form factors of the unperturbed Hamiltonian [56]. One interesting phenomenon to potentially study with this approach is Floquet prethermalization, as has been discussed in [57,58], observed for driving protocols that only break integrability weakly.

## Acknowledgements

I thank Jean-Sebastien Caux, Vladimir Gritsev, Neil J. Robinson and Dirk Schuricht for discussions and helpful comments on this manuscript. This work is supported by the European Union's Horizon 2020 under the Marie Sklodowoska-Curie grant agreement 750092.

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
