# Peer review of "Integrable Floquet QFT: Elasticity and factorization under periodic driving"

_SciPost Physics, doi:SciPost Phys. 5, 025 (2018)_

## Round 1 · Referee Report · Anonymous (Referee 2) · 2018-6-15

Strengths

1- Axiomatic definition of integrability for Floquet systems 2-Manuscript well-written and easy to follow

Weaknesses

1- The examples are mostly non-interacting and not especially exciting from a physics perspective.

Report

In this paper, the author introduces a definition of integrability for periodically driven (Floquet) systems. Whereas the idea of Floquet integrability was proposed before, this is the first concrete and complete definition of what integrability could mean for such systems. The author follows an axiomatic approach that could be useful to identify and analyze integrable Floquet systems. Examples and counter examples of integrability and quasi-integrability are given, mostly for non-interacting theories. Overall I think this paper is interesting and it is certainly satisfying to have a concrete framework for Floquet integrability. My main concern with this work is that it remains unclear whether there exists any interacting, non-trivial example of Floquet integrable system. This is an important issue that should be discussed in the paper. The only interacting example uses the boost operator and already appeared in a previous work by Gritsev and Polkovnikov (and clearly such protocols are quite artificial). On the other hand, it remains very unclear to me that a non-trivial example like eq 53 is integrable in any meaningful sense. Even if it is hard to show integrability in the way introduced by the author, I think some arguments to explain why this model could be integrable are warranted. Could the author exhibit at least one or two local conserved quantities for this model? A thorough discussion of interacting examples would make this framework much more exciting.

Requested changes

1- Provide a concrete path to find interacting examples not based on the boost operator 2- Extend the discussion of interacting models, especially the XXX chain since this is the most interesting example.

---

## Round 1 · Referee Report · Anonymous (Referee 1) · 2018-6-15

Strengths

1-Axiomatic construction of integrable quantum field theory under periodic driving based on general principles in the style of the S-matrix bootstrap program.

2-Nice presentation, including clear introduction to the standard S-matrix bootstrap program and detailed calculations.

Weaknesses

1-Definition of Floquet integrability based on infinite set of local conserved charges may be too restricting for practical uses.

2-Examples are mostly trivial, approximate and not related to many-body interactions.

Report

In this work the author discusses aspects of integrability in periodically-driven quantum field theories. Following the reasoning of standard integrable QFT, the author proposes a definition of Floquet integrability associated with existence of an infinite set of local charges that commute with the Floquet Hamiltonian (the effective Hamiltonian describing the time-evolution of the system over each driving period). In analogy to the standard scattering theory arguments on integrability and the S-matrix bootstrap program, the author then argues that Floquet integrability guarantees elasticity and factorisation of particle collisions under stroboscopic observations of the system which imply certain constraints on the stroboscopic time evolution. The rest of this work focuses on examples of driving protocols: it is first shown that the simple choice of periodic mass quenches in free bosonic or fermionic field theories does not satisfy the conditions of Floquet integrability and later on some trivial cases of approximate Floquet integrability are discussed.

Overall this work introduces an extension of concepts and ideas of integrable QFT to the case of periodic driving that are useful for an axiomatic construction of stroboscopically integrable QFT. Motivation for the study of periodic driving dynamics comes from quantum or condensed matter physics and experimental applications for which quantum field theories are useful as effective descriptions at least at equilibrium or close to it. However the relativistic invariance of QFT, which is crucial in the S-matrix bootstrap program, is only approximate in condensed matter or experimental applications and so the dynamics under periodic driving would diverge from the QFT predictions rather quickly. This fact restricts somehow the potential use of these results in this direction. Moreover the definition of Floquet integrability introduced here, even though directly analogous to the standard one, may be too strong to describe any example of practical interest. For example even the simple protocol of free field theory periodic driving that is exactly solvable due to the closeness of the algebra of quadratic operators and which does not lead to infinite heating up, is excluded from this definition of Floquet integrability. Almost all examples discussed later are rather trivial cases of approximate Floquet integrability that would work even for a small quantum system or are based on very exceptional fine-tuning that reduces the dynamics again to a quantum mechanical rather than statistical physics problem (exact revivals or fully degenerate energy spectrum). Perhaps the only non-trivial case is the boost-operator protocol of sec. 6.3.2.

Therefore one conclusion of this work could have been that defining Floquet integrability in QFT by requiring existence of an infinite set of local conserved charges, in direct analogy to the standard case, is too constraining, which is certainly an interesting observation.

Requested changes

In section 4, the author introduces a function F that describes the time evolution of a single-particle state over one driving period and determines the analytical properties that it must satisfy. This is essentially the equivalent of the travelling phase factor. Question: shouldn’t one allow also for an independent S-matrix describing the elastic collisions during one driving period? In sec.4 collisions seem to be allowed only during the part of the driving steps when time-evolution follows $H_{int}$ and so the corresponding S-matrix is the standard one. How would the axioms be modified if one took into account this possibility?

p.12 explain better how the requirement of rapidity set conservation after n periods implies the same after each one period separately

p.12 “If we consider a large enough number of periods, n, the particles in the final state (12), can be considered to be well separated from each other. This assumption that the final state can be represented in terms of well separated asymptotic particle states is sensible in translationally invariant systems” I think translational invariance is not sufficient to guarantee well-separation of outgoing particles. Locality of the Floquet Hamiltonian seems to be necessary. It should anyway be clarified if the Floquet Hamiltonian is assumed to be local or if the infinite set of conserved charges is assumed to be local without the Floquet Hamiltonian being such.

Minor changes:

  • I think this paper doesn't really fit in the subject area "Condensed Matter Physics - Theory" but rather in "high energy".

  • eq. (12): sum over i must be a typo

  • it would be good to cite P. Dorey’s “exact S-matrices” or refs. therein as a reference on the standard scattering theory arguments relating presence of local conserved charges to factorisability, elasticity etc. that have been used in this paper.

  • Shouldn’t F(\theta) be unitary?

  • p.22: “Suppose for example, that we choose the initial state to be a one-particle eigenstate…” I think the requirement that the Bogolyubov coefficient d should vanish needs at least a two-particle state to demonstrate (because d is the coefficient of the annihilation operator which annihilates the vacuum).

  • “For a general system a full revival is expected when the system is placed in a finite volume, however, the time when a revival is expected grows exponentially with system size…” It’s not accurate to say that exact revivals are possible in general systems but correspond to exponentially large revival times. For a general system exact revivals are just impossible (exact revivals require either all energy levels being multiples of a fundamental frequency as in CFT or fine-tuning). In general systems the dynamics is quasi-periodic meaning that revivals are only approximate (in the sense of the quantum recurrence theorem mentioned in 6.2.1).

  • p.27: “Bogoliubov transformation describing each step of the driving, as d(2),p = 0, for all momenta, p, and integer n.”: erase “integer n” since there isn’t here.

-(47) and (48) can be simplified further: e.g. $E_{1,p}^2-E_{2,p}^2$ is just $m_1^2-m_2^2$.

  • p.31: isn’t this protocol the same as the Onsager algebra protocol of [13] discussed later? so that the Floquet Hamiltonian can be found from the BCH expansion as linear combination of the generators (of course this doesn’t mean it has a trivial form)

  • p.33: notation (3^n) and Z[0] not so clear in the unnumbered eq. defining U,V. The meaning of parity is also unclear (in terms of the fermionic fields?)

---

## Round 2 · Referee Report · Anonymous · 2018-9-1

Report

I have read the author’s response and the revised manuscript. I found the response and the changes made satisfactory, and I'm therefore recommending publication in SciPost.

---

## Round 2 · Referee Report · Anonymous · 2018-9-5

Report

In this new version the author has addressed my questions and has sufficiently extended/clarified the discussion on the possibility of finding driving protocols satisfying his definition of Floquet integrability. Even though the analysis is not conclusive I appreciate that at this stage it is hard to give a definite answer and this work should serve as a first step towards more in-depth investigations, therefore I recommend it for publication.

---

## Round 2 · Author Response

I first want to thank the referees for their time and useful comments. I have incorporated their suggestions into this new version of the manuscript, which I hope they will find more suitable. I list below the changes made to the manuscript, in reply to the requests from the referees.

Referee report 1:

I first remark, that it is true that at present, the examples of Floquet integrability discussed are very simple, and it would be much more interesting in the future to see if some more intricate and truly interacting integrable protocols can be found. Nevertheless, there does not seem to be any particular “no-go theorem” prohibiting the discovery of more interesting examples, and I list throughout the manuscript several promising future directions to explore. I therefore don’t really agree with the suggested conclusion that integrability is “too constraining”, since I do think it will soon be possible to find other interesting examples.

-I clarified the discussion on the possibility two particles scattering within the “shaded region” governed by $H’(t)$. In general, this would be described by some combined phase, $F(\theta_1,\theta_2)$. I argue, that as a consequence of stroboscopic factorizability, it must be possible to express this function in terms of the one-particle phase, $F(\theta)$, and the standard two-particle S-matrix. I extended the discussion of the Floquet Yang-Baxter equation, to include this. This is similar to the case of three-particle scattering in standard equilibrium integrable QFT, where the general three-particle S-matrix can be reduced in terms of products of 2-particle S-matrices.

-The assumption that the set of rapidities are conserved after each period, is just the simplest solution to the requirement that rapidities are conserved after $n$-periods. The beginning of Section 4 has been slightly rewritten to make this more clear. This is similar to the standard equilibrium case, where one can only directly derive the fact that the set of rapidities is conserved between the asymptotic in and out states, yet the simplest assumption is that the set of particle rapidities is conserved at intermediate times as well.

-We are not requiring locality of the Floquet Hamiltonian, however, we require that it commutes with some local conserved charges. The discussion regarding “well separated particles” has been modified and made clearer in the text. The only requirements are that the basis of asymptotic particle states is complete and spans the Hilbert space, such that the state after $n$ periods can be expressed in terms of particle states, and then we require that the conserved charges be local. From these requirements, Eq. (13) follows, and it is not necessary that the Floquet Hamiltonian be local.

-I have now changed the main subject to High Energy Theory, and kept Condensed Matter as a secondary subject. I think the subject of the manuscript lies somewhere in between, and it is hard to classify, and perhaps may be of interest to both communities. I would leave the final decision up to the editor in charge in any case.

-Indeed the sum over $i$ in (12) was a mistake and has been corrected now.

-I have now included a reference to P. Dorey at the beginning of Section, I agree it is a relevant review.

-The condition of unitarity on $F(\theta)$ is mainly what I call the Floquet Annihilation axiom. This is the statement that $F(\theta)$ can be cancelled against the opposite phase, arising from an antiparticle, which is the manifestation of unitarity in the $F(\theta)$ function.

-p.22: “Suppose for example, that we choose the initial state to be a one-particle eigenstate…” I agree it is necessary to make the argument with a multi particle state, that the Bogoliubov coefficient d should vanish, and this was a mistake in the previous version. It has now been corrected with a multi particle state.

-I clarified the issue of revivals in generic systems, it is made clear that in general one can only expect approximate revivals, and I have made sure to point this out where relevant in the manuscript.

-p.27: “Bogoliubov transformation describing each step of the driving, as d(2),p = 0, for all momenta, p, and integer n.” I have corrected this typo, now it reads $d_{(2n),p}=0$

-Equations (47) and (48) (now 48 and 49 in this version) have been simplified, inserting the dispersion relations.

-The protocol described in 6.2.3 is not exactly the same as the Onsager protocol described in [13]. The Onsager protocol consists of alternating between the Hamiltonians $A_0$ and $A_1$ from Eq. (53). The protocol in 6.2.3 consists of a two-step driving process where one of the Hamiltonians is either $A_0$ or $A_1$, but the second Hamiltonian can be a full transverse field Ising chain, involving both terms, So it is in this sense more general than the Onsager protocol. The Protocol of 6.2.3 is initially expected to be integrable only when tuned to the appropriate revival time. The Onsager protocol of [13], being simpler, was proposed to be integrable for any driving period, though we find that it doesn’t fulfill all our requirements of integrability except at the fine-tuned revival times.

-The notation (3^n) and Z[0], was taken from the original reference [27], but I have now made more transparent in the text what this means. The references to parity were also modified and made clearer in this section. Conserved charges built out of only the U operators are invariant under spatial inversion, $x\to-x$.

Referee report 2:

-I have expanded the discussion in the Conclusions section about how other less trivial integrable Floquet systems may be found. We point out that recently a protocol very similar to the “quantum dilatation clock” proposed in the conclusions was studied, and the results seem to be compatible with our definition of integrability, but this has to be studied further. I also extended the discussion on the study of Floquet critical phenomena, and outlined clearly the steps that would need to be followed to show any connection between critical phenomena and integrability in 2d driven systems.

-I have added a paragraph at the end of 6.3.1, discussing the XXX protocol proposed in [13]. While we are not able at present to study the properties of this protocol as we did with those related to free systems, there is reason to be hopeful that it may agree with our definition of integrability. As we discuss, the first few terms in the BCH expansion of the Floquet Hamiltonian do seem to commute with local, and parity-even conserved charges, so the protocol seems to be integrable at least at this perturbative level. This is not true for the generic Onsager protocol also discussed in this section, which does not commute with parity-even charges.

In general, as I replied to the first referee, there does not seem to be any “no-go theorem” preventing more interesting interacting integrable protocols from being discovered in the future, and there are possible open pathways to discover new protocols. So the fact that at this point we can only discuss very simple protocols in detail should not mean that only these protocols are possible.

---

## Editorial Decision

published